# Tuning Frequency Bias of State Space Models

**Annan Yu**[1] **Dongwei Lyu**[2] **Soon Hoe Lim**[3,4] **Michael W. Mahoney**[5–7] **N. Benjamin Erichson**[5,6]

[1] Center for Applied Mathematics, Cornell University, Ithaca, NY 14853, USA

[2] Data Science Institute, University of Chicago, Chicago, IL 60637, USA

[3] Department of Mathematics, KTH Royal Institute of Technology, Stockholm, Sweden

[4] Nordita, KTH Royal Institute of Technology and Stockholm University, Stockholm, Sweden

[5] Lawrence Berkeley National Laboratory, Berkeley, CA 94720, USA

[6] International Computer Science Institute, Berkeley, CA 94704, USA

[7] Department of Statistics, University of California at Berkeley, Berkeley, CA 94720, USA

`ay262@cornell.edu`, `dwlyu@uchicago.edu`, `shlim@kth.se`, `mmahoney@stat.berkeley.edu`, `erichson@icsi.berkeley.edu`

## Abstract

State space models (SSMs) leverage linear, time-invariant (LTI) systems to effectively learn sequences with long-range dependencies. By analyzing the transfer functions of LTI systems, we find that SSMs exhibit an implicit bias toward capturing low-frequency components more effectively than high-frequency ones. This behavior aligns with the broader notion of frequency bias in deep learning model training. We show that the initialization of an SSM assigns it an innate frequency bias and that training the model in a conventional way does not alter this bias. Based on our theory, we propose two mechanisms to tune frequency bias: either by scaling the initialization to tune the inborn frequency bias; or by applying a Sobolev-norm-based filter to adjust the sensitivity of the gradients to high-frequency inputs, which allows us to change the frequency bias via training. Using an image-denoising task, we empirically show that we can strengthen, weaken, or even reverse the frequency bias using both mechanisms. By tuning the frequency bias, we can also improve SSMs' performance on learning long-range sequences, averaging an $88.26\%$ accuracy on the Long-Range Arena (LRA) benchmark tasks.

## 1 Introduction

Sequential data are ubiquitous in fields such as natural language processing, computer vision, generative modeling, and scientific machine learning. Numerous specialized classes of sequential models have been developed, including recurrent neural networks (RNNs) (Arjovsky et al., 2016; Chang et al., 2019; Erichson et al., 2021; Rusch & Mishra, 2021; Orvieto et al., 2023), convolutional neural networks (CNNs) (Bai et al., 2018; Romero et al., 2022), continuous-time models (CTMs) (Gu et al., 2021b; Yildiz et al., 2021), transformers (Katharopoulos et al., 2020; Choromanski et al., 2020; Kitaev et al., 2020; Zhou et al., 2022; Nie et al., 2023), state space models (SSMs) (Gu et al., 2022b;a; Hasani et al., 2023; Smith et al., 2023), and Mamba (Gu & Dao, 2023; Dao & Gu, 2024). Among these, SSMs stand out for their ability to learn sequences with long-range dependencies.

Using the continuous-time linear, time-invariant (LTI) systems,

$$\mathbf{x}'(t) = \mathbf{A}\mathbf{x}(t) + \mathbf{B}\mathbf{u}(t), \qquad \mathbf{y}(t) = \mathbf{C}\mathbf{x}(t) + \mathbf{D}\mathbf{u}(t), \tag{1}$$

where $\mathbf{A} \in \mathbb{C}^{n \times n}$, $\mathbf{B} \in \mathbb{C}^{n \times m}$, $\mathbf{C} \in \mathbb{C}^{p \times n}$, and $\mathbf{D} \in \mathbb{C}^{p \times m}$, an SSM computes the output time-series $\mathbf{y}(t)$ from the input $\mathbf{u}(t)$ via a latent state vector $\mathbf{x}(t)$. Compared to an RNN, a major computational advantage of an SSM is that the LTI system can be trained both efficiently (i.e., the training can be parallelized for long sequences) and numerically robustly (i.e., it does not suffer from vanishing and exploding gradients). An LTI system can be computed in the time domain via convolution:

$$\mathbf{y}(t) = (\mathbf{h} * \mathbf{u} + \mathbf{D}\mathbf{u})(t) = \int_{-\infty}^{\infty} \mathbf{h}(t - \tau)\mathbf{u}(\tau)d\tau + \mathbf{D}\mathbf{u}(t), \qquad \mathbf{h}(t) = \mathbf{C}\exp(t\mathbf{A})\mathbf{B}.$$

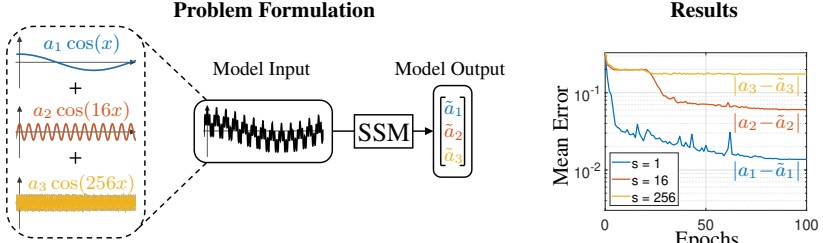

**Figure 1:** In a synthetic example to illustrate the frequency bias of SSMs, we form the inputs by superposing three waves of low, moderate, and high frequencies, respectively. We train an S4D model to regress the magnitudes of the three waves. We observe that the magnitudes of the low-frequency waves can be approximated much better compared to those of the high-frequency waves. In Figure 10, we show how to tune the frequency bias in this example.

Alternatively, it can be viewed as an action in the frequency domain:

$$\hat{\mathbf{y}}(s) = \mathbf{G}(is)\hat{\mathbf{u}}(s), \qquad \mathbf{G}(is) := \mathbf{C}(is\mathbf{I} - \mathbf{A})^{-1}\mathbf{B} + \mathbf{D}, \quad s \in \mathbb{R}, \tag{2}$$

where $i$ is the imaginary unit and $\mathbf{I}$ is the identity matrix. The function $\mathbf{G}$ is called the transfer function of the LTI system.

The frequency-domain characterization of the LTI systems in eq. (2) sets the stage for understanding the so-called frequency bias of an SSM. The term "frequency bias" originated from the study of a general overparameterized multilayer perceptron (MLP) (Rahaman et al., 2019), where it was observed that the low-frequency content was learned much faster than the high-frequency content. It is a form of implicit regularization (Mahoney, 2012). Frequency bias is a double-edged sword: on one hand, it partially explains the good generalization capability of deep learning models, because most high-frequency noises are not learned until the low-frequency components are well-captured; on the other hand, it puts a curse on learning the useful high-frequency information in the target.

In this paper, we aim to understand the frequency bias of SSMs. In Figure 1, we observe that, similar to most deep learning models, SSMs are also better at learning the low frequencies than the high ones. To understand that, we develop a theory that connects the spectrum of $\mathbf{A}$ to the SSM's capability of processing high-frequency signals. Then, based on the spectrum of $\mathbf{A}$, we analyze the frequency bias in two steps. First, we show that the most popular initialization schemes (Gu et al., 2020; 2021b; Yu et al., 2024b) lead to SSMs that have an innate frequency bias. More precisely, they place the spectrum of $\mathbf{A}$, $\Lambda(\mathbf{A})$, in the low-frequency region in the $s$-plane, preventing LTI systems from processing high-frequency input, regardless of the values of $\mathbf{B}$ and $\mathbf{C}$. Second, we consider the training of the SSMs. Using the decay properties of the transfer function, we show that if an eigenvalue $a_j \in \Lambda(\mathbf{A})$ is initialized in the low-frequency region, then its gradient is insensitive to the loss induced by the high-frequency input content. *Hence, if an SSM is not initialized with the capability of handling high-frequency inputs, then it will not be trained to do so by conventional training.*

The initialization of the LTI systems equips an SSM with a certain frequency bias, but this is not necessarily the appropriate implicit bias for a given task. Depending on whether an SSM needs more expressiveness or generalizability, we may want less or more frequency bias, respectively (see Figure 10). Motivated by our analysis, we propose two ways to tune the frequency bias:

1. Instead of using the HiPPO initializations, the most popular class of initializations used in practice, we scale $\Lambda(\mathbf{A})$ to lower or higher-frequency regions at initialization as a "hard tuning strategy" that marks out the regions in the frequency domain that can be learned.
2. Motivated by the Sobolev norm, which applies weights to the Fourier domain, we can apply a multiplicative factor of $(1 + |s|)^\beta$ to the transfer function $\mathbf{G}(is)$. This is a "soft tuning strategy" that reweighs each location in the frequency domain. By selecting a positive or negative $\beta$, we make the gradients more or even less sensitive to the high-frequency input content, respectively, which changes the frequency bias during training.

One can think of these two mechanisms as ways to tune frequency bias at initialization and during training, respectively. After rigorously analyzing them, we present an experiment on image-denoising with different noise frequencies to demonstrate their effectiveness. We also show that tuning the frequency bias enables better performance on tasks involving long-range sequences. Equipped with our two tuning strategies, a simple S4D model can be trained to average an $88.26\%$ accuracy on the Long-Range Arena (LRA) benchmark tasks (Tay et al., 2021).

**Contribution.** Here are our main contributions: (1) We formalize the notion of frequency bias for SSMs and quantify it using the spectrum of $\mathbf{A}$. We show that a diagonal SSM initialized by HiPPO has an innate frequency bias. We are the first to study the training of the state matrix $\mathbf{A}$, and we show that training the SSM does not alter this frequency bias. (2) We propose two ways to tune frequency bias, by scaling the initialization, and by applying a Sobolev-norm-based filter to the transfer function of the LTI systems. We study the theory of both strategies and provide guidelines for using them in practice. (3) We empirically demonstrate the effectiveness of our tuning strategies using an image-denoising task. We also show that tuning the frequency bias helps an S4D model to achieve state-of-the-art performance on the Long-Range Arena tasks and provide ablation studies.

To make the presentation cleaner, throughout this paper, we focus on a single *single-input/single-output (SISO)* LTI system $\Gamma = (\mathbf{A}, \mathbf{B}, \mathbf{C}, \mathbf{D})$ in an SSM, i.e., $m = p = 1$, although all discussions naturally extend to the multiple-input/multiple-output (MIMO) case, as in an S5 model (Smith et al., 2023). Hence, the transfer function $\mathbf{G} : \mathbb{C} \to \mathbb{C}$ is complex-valued. We emphasize that while we focus on a single system $\Gamma$, we do *not* isolate it from a large SSM; in fact, when we study the training of $\Gamma$ in section 4, we backpropagate through the entire SSM.

**Related Work.** The frequency bias, also known as the spectral bias, of a general neural network (NN) was initially observed and studied in Rahaman et al. (2019); Yang & Salman (2019); Xu (2020). The name spectral bias stemmed from the spectral decomposition of the so-called neural tangent kernels (NTKs) (Jacot et al., 2018), which provides a means of approximating the training dynamics of an overparameterized NN (Arora et al., 2019; Su & Yang, 2019; Cao et al., 2019). By carefully analyzing the eigenfunctions of the NTKs, Basri et al. (2019); Bietti & Mairal (2019) proved the frequency bias of an overparameterized two-layer NN for uniform input data. The case of nonuniform input data was later studied in Basri et al. (2020); Yu et al. (2023). The idea of Sobolev-norm-based training of NNs has been considered in Vlassis & Sun (2021); Yu et al. (2023); Tsay (2021); Son et al. (2021); Czarnecki et al. (2017); Zhu et al. (2021); Son (2023); Liu et al. (2024). The initialization of the LTI systems in SSMs plays a crucial role, which was first observed in Gu et al. (2020). Empirically successful initialization schemes called "HiPPO" were proposed in Voelker et al. (2019); Gu et al. (2020; 2023). Other efforts in improving the initialization of an SSM were studied in Yu et al. (2024b); Liu & Li (2024a). Later, Orvieto et al. (2023); Yu et al. (2024a) attributed the success of HiPPO to the proximity of the spectrum of $\mathbf{A}$ to the imaginary axis (i.e., the *real* parts of the eigenvalues of $\mathbf{A}$ are close to zero). This paper considers the *imaginary* parts of the eigenvalues of $\mathbf{A}$, which was also discussed in the context of the approximation-estimation tradeoff in Liu & Li (2024b). The training of SSMs has mainly been considered in Smékal et al. (2024); Liu & Li (2024a), where the matrix $\mathbf{A}$ is assumed to be fixed, making the optimization convex. To our knowledge, we are the first to consider the training of $\mathbf{A}$. While we consider the decay of the transfer functions of the LTI systems in the frequency domain, there is extensive literature on the decay of the convolutional kernels in the time domain (i.e., the memory) (Hardt et al., 2018; Gu et al., 2020; Wang & Li, 2023; Wang & Xue, 2024; Orvieto et al., 2024; Yu et al., 2024a).

## 2 WHAT IS THE FREQUENCY BIAS OF AN SSM?

In Figure 1, we see an example where an S4D model is better at predicting the magnitude of a low-frequency component in the input than a high-frequency one. This coincides with our intuitive interpretation of frequency bias: the model is better at "handling" low frequencies than high frequencies. To rigorously analyze this phenomenon for SSMs, however, we need to formalize the notion of frequency bias. This is our goal in this section. One might imagine that an SSM has a frequency bias if, given a time-series input $\mathbf{u}(t)$ that has rich high-frequency information, its time-series output $\mathbf{y}(t)$ lacks high-frequency content. Unfortunately, this is not the case: an SSM is capable of generating high-frequency outputs. Indeed, the skip connection $\mathbf{D}$ of an LTI system

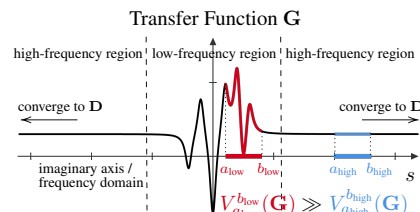

**Figure 2:** The frequency bias of an SSM says that the frequency response has more variation in the low-frequency area than the high-frequency one.

is an "all-pass" filter, multiplying the whole input $\mathbf{u}(t)$ by a factor of $\mathbf{D}$ and adding it to the output $\mathbf{y}(t)$. On the other hand, the secret of a successful SSM hides in $\mathbf{A}$, $\mathbf{B}$, and $\mathbf{C}$ (Yu et al., 2024a).

In an ablation study, when $\mathbf{D}$ is removed, an S4D model only loses less than $2\%$ of accuracy on the sCIFAR-10 task (Tay et al., 2021; Krizhevsky et al., 2009), whereas the model completely fails when we remove $\mathbf{A}$, $\mathbf{B}$, and $\mathbf{C}$. This can be ascribed to the LTI system's power to model complicated behaviors in the frequency domain. That is, each Fourier mode in the input has its own distinct "pass rate" (see Adamyan et al. (1971); Sun (2020); Yu & Townsend (2024) for why this is an important feature of LTI systems). For example, the task in Figure 1 can be trivially solved if the LTI system can filter out a single mode $a_1 \cos(x)$, $a_2 \cos(16x)$, or $a_3 \cos(256x)$ from the superposition of the three; the skip connection $\mathbf{D}$ alone is not capable of doing that.

Given that, we can formulate the frequency bias of an LTI system as follows (see Figure 2):

*Frequency bias of an SSM means that the frequency responses (i.e., the transfer functions $\mathbf{G}$) of LTI systems have more variation in the low-frequency area than the high-frequency area.*

More precisely, given the transfer function $\mathbf{G}(is)$, we can study its total variation in a particular interval $[a, b]$ in the Fourier domain defined by

$$V_a^b(\mathbf{G}) := \sup_{\substack{a=s_0 < s_1 < \cdots < s_N = b, \\ N \in \mathbb{N}}} \sum_{j=1}^N |\mathbf{G}(is_j) - \mathbf{G}(is_{j-1})| = \int_a^b \left| \frac{d\mathbf{G}(is)}{ds} \right| ds, \quad -\infty \le a < b \le \infty.$$

Intuitively, $V_a^b(\mathbf{G})$ measures the total change of $\mathbf{G}(is)$ when $s$ moves from $a$ to $b$. The larger it is, the better an LTI system is at distinguishing the Fourier modes with frequencies between $a$ and $b$. Frequency bias thus says that for a fixed-length interval $[a, b]$, $V_a^b(\mathbf{G})$ is larger when $[a, b]$ is near the origin than when it lies in the high-frequency region, i.e., when it is far from the origin.

## 3 FREQUENCY BIAS OF AN SSM AT INITIALIZATION

Our exploration of the frequency bias of an SSM starts with the initialization of a SISO LTI system $(\mathbf{A}, \mathbf{B}, \mathbf{C}, \mathbf{D})$, where $\mathbf{A} = \mathrm{diag}(a_1, \ldots, a_n) \in \mathbb{C}^{n \times n}$ is diagonal. The system is assumed to be stable, meaning that $a_j = v_j + iw_j$ for some $v_j < 0$ for all $1 \le j \le n$, where $v_j$ and $w_j$ are the real and the imaginary parts of $a_j$, respectively. Note that the diagonal structure of $\mathbf{A}$ is indeed the most popular choice for SSMs (Gu et al., 2022a; Smith et al., 2023), in which case it suffices to consider the Hadamard (i.e., entrywise) product $\mathbf{B} \circ \mathbf{C}^\top = [c_1 \quad \cdots \quad c_n]^\top \in \mathbb{C}^n$, where $c_j = \xi_j + i\zeta_j$ for all $1 \le j \le n$. Then, the transfer function $\mathbf{G}$ is naturally represented in partial fractions:

$$\mathbf{G}(is) = \frac{c_1}{is - a_1} + \cdots + \frac{c_n}{is - a_n} + \mathbf{D} = \frac{\xi_1 + i\zeta_1}{-v_1 + i(s - w_1)} + \cdots + \frac{\xi_n + i\zeta_n}{-v_n + i(s - w_n)} + \mathbf{D}.$$

In most cases, the input $\mathbf{u}(t)$ is real-valued. To ensure that output is real-valued, the standard practice is to take the real part of $\mathbf{G}$ as a real-valued transfer function before applying eq. (2):

$$\tilde{\mathbf{G}}(is) := \mathrm{Re}(\mathbf{G}(is)) = \sum_{j=1}^n \frac{\zeta_j(s - w_j) - \xi_j v_j}{v_j^2 + (s - w_j)^2} + \mathbf{D}. \tag{3}$$

We now derive a general statement for the total variation of $\tilde{\mathbf{G}}$ given the distribution of $w_j$.

**Lemma 1.** Let $\tilde{\mathbf{G}}$ be the transfer function defined in eq. (3). Given any $B > \max_j |w_j|$, we have

$$V_{-\infty}^{-B}(\tilde{\mathbf{G}}) \le \sum_{j=1}^n \frac{|c_j|}{|w_j + B|}, \qquad V_B^\infty(\tilde{\mathbf{G}}) \le \sum_{j=1}^n \frac{|c_j|}{|w_j - B|}.$$

Given the formula of the transfer function, the proof of Lemma 1 is almost immediate. In this paper, we leave all proofs to Appendix B and D. Lemma 1 illustrates a clear and intuitive concept:

*If the imaginary parts of $a_j$ are distributed in the low-frequency region, i.e., $|w_j|$ are small, the transfer function has a small total variation in the high-frequency areas $(-\infty, -B]$ and $[B, \infty)$ as $B \to \infty$, inducing a frequency bias of the SSM.*

We can now apply Lemma 1 to study the innate frequency bias of the HiPPO initialization (Gu et al., 2020). While there are many variants of HiPPO, we choose the one that is commonly used in practice (Gu et al., 2021a). All other variants can be similarly analyzed.

**Corollary 1.** Assume that $a_j = -0.5 + i(-1)^j \lfloor j/2 \rfloor \pi$ and $\xi_j, \zeta_j \sim \mathcal{N}(0,1)$ i.i.d., where $\mathcal{N}(0,1)$ is the standard normal distribution. Then, given $B > n\pi/2$ and $\delta > 0$, we have

$$V_{-\infty}^{-B}(\tilde{\mathbf{G}}), V_B^{\infty}(\tilde{\mathbf{G}}) \leq \frac{\sqrt{2n}(\sqrt{n} + \sqrt{\ln(1/\delta)})}{B - n/2} \qquad \text{with probability} \geq 1 - \delta.$$

In particular, Corollary 1 tells us that the HiPPO initialization only captures the frequencies $s \in [-B, B]$ up to $B = \mathcal{O}(n)$, because when $B = \omega(n)$, we see that $V_{-\infty}^{-B}(\tilde{\mathbf{G}}), V_B^{\infty}(\tilde{\mathbf{G}})$ vanish as $n$ increases. This means that no complicated high-frequency responses can be learned.

## 4 FREQUENCY BIAS OF AN SSM DURING TRAINING

In section 3, we see that the initialization of the LTI systems equips an SSM with an innate frequency bias. A natural question to ask is whether an SSM can be *trained* to adopt high-frequency responses. Analyzing the training of an SSM (or many other deep learning models) is not an easy task, and we lack theoretical characterizations. Two notable exceptions are Liu & Li (2024b); Smékal et al. (2024), where the convergence of a trainable LTI system to a target LTI system is analyzed, assuming that the state matrix $\mathbf{A}$ is fixed to make the optimization problem convex. Unfortunately, this assumption is too strong to be applied for our purpose. Indeed, Lemma 1 characterizes the frequency bias using the distribution of $w_j = \text{Im}(a_j)$, making the training dynamics of $\mathbf{A}$ a crucial element in our analysis. Even if we set aside the issue of $\mathbf{A}$, analyzing an isolated LTI system in an SSM remains unrealistic: when an SSM, consisting of hundreds of LTI systems, is trained for a single task, there is no clear notion of "ground truth" for each individual LTI system within the model.

To make our discussion truly generic, we assume that there is a loss function $\mathcal{L}(\boldsymbol{\Theta})$ that depends on all parameters $\boldsymbol{\Theta}$ of an SSM. In particular, $\boldsymbol{\Theta}$ contains $v_j, w_j, \xi_j, \zeta_j$, and $\mathbf{D}$ from every LTI system within the SSM, as well as the encoder, decoder, and inter-layer connections. With mild assumptions on the regularity of the loss function $\mathcal{L}$, we provide a quantification of the gradient of $\mathcal{L}$ with respect to $w_j$ that leads to a qualitative statement about the frequency bias during training.

**Theorem 1.** Let $\mathcal{L}(\boldsymbol{\Theta})$ be a loss function and $(\mathbf{A}, \mathbf{B}, \mathbf{C}, \mathbf{D})$ be a diagonal LTI system in an SSM defined in section 3. Let $\tilde{\mathbf{G}}$ be its associated real-valued transfer function defined in eq. (3). Suppose the functional derivative of $\mathcal{L}(\boldsymbol{\Theta})$ with respect to $\tilde{\mathbf{G}}(is)$ exists and is denoted by $(\partial/\partial\tilde{\mathbf{G}}(is))\mathcal{L}$. Then, if $|(\partial/\partial\tilde{\mathbf{G}}(is))\mathcal{L}| = \mathcal{O}(|s|^p)$ for some $p < 1$, we have

$$\frac{\partial \mathcal{L}}{\partial w_j} = \int_{-\infty}^{\infty} \frac{\partial \mathcal{L}}{\partial \tilde{\mathbf{G}}(is)} \cdot K_j(s) \, ds, \qquad K_j(s) := \frac{\zeta_j((s - w_j)^2 - v_j^2) - 2\xi_j v_j(s - w_j)}{[v_j^2 + (s - w_j)^2]^2}, \quad (4)$$

for every $1 \leq j \leq n$. In particular, we have that $|K_j(s)| = \mathcal{O}\left(|\zeta_j s^{-2}| + |\xi_j s^{-3}|\right)$ as $|s| \to \infty$.

In Theorem 1, we use a technical tool called the functional derivative (Gelfand et al., 2000). The assumption that $(\partial/\partial\tilde{\mathbf{G}}(is))\mathcal{L}$ exists is easily satisfied, and we leave a survey of functional derivatives to Appendix C. The assumption that $|(\partial/\partial\tilde{\mathbf{G}}(is))\mathcal{L}|$ grows at most sublinearly is to guarantee the convergence of the integral in eq. (4); it is also easily satisfiable. We will see that the growth/decay rate of $|(\partial/\partial\tilde{\mathbf{G}}(is))\mathcal{L}|$ plays a more important role when we start to tune the frequency bias using the Sobolev-norm-based method (see section 5.2). As usual, one can intuitively think of the functional derivative $(\partial/\partial\tilde{\mathbf{G}}(is))\mathcal{L}$ as a measurement of the "sensitivity" of the loss function $\mathcal{L}$ to an LTI system's action on a particular frequency $s$ (i.e., $\tilde{\mathbf{G}}(is)$). The fact that it is multiplied by a factor of $K_j(s)$ in the computation of the gradient in eq. (4) conveys the following important message:

> *The gradient of $\mathcal{L}$ with respect to $w_j$ highly depends on the part of the loss that has "local" frequencies near $s = w_j$. It is relatively unresponsive to the loss induced by high frequencies, with a decaying factor of $\mathcal{O}(|s|^{-2})$ as the frequency increases, i.e., as $|s| \to \infty$.*

Hence, the loss landscape of the frequency domain contains many local minima, and an LTI system can rarely learn the high frequencies with the usual training. To verify this, we train an S4D model initialized by HiPPO to learn the sCIFAR-10 task for 100 epochs. We measure the relative change of each parameter $\theta$: $\Delta(\boldsymbol{\theta}) = (|\theta^{(0)} - \theta^{(100)}|)/(|\theta^{(0)}|)$, where the superscripts indicate the epoch number. As we will show in section 6, the HiPPO initialization is unable to capture the high frequencies in the CIFAR-10 pictures fully. From Table 1, however, we see that $\text{Im}(\text{diag}(\mathbf{A}))$ is trained

very little: every $w_j$ is only shifted by $1.43\%$ on average. This can be explained by Theorem 1: $w_j$ is easily trapped by a low-frequency local minimum.

**Table 1:** The average relative change of each LTI system matrix in an S4D model trained on the sCIFAR-10 task. We see that the imaginary parts of $\mathrm{diag}(\mathbf{A})$ are almost unchanged during training.

| Parameter | Re(diag($\mathbf{A}$)) | Im(diag($\mathbf{A}$)) | $\mathbf{B} \circ \mathbf{C}^\top$ | $\mathbf{D}$ |
|---|---|---|---|---|
| $\Delta$ | 1002.705 | 0.0143 | 1.1801 | 0.8913 |

**An Illustrative Example.** Our analysis of the training dynamics of $w_j$ in Theorem 1 is very generic, relying on the notion of the functional derivatives. To make the theorem more concrete, we consider a synthetic example (see Figure 3). We fall back to the case of approximating a target function

$$\tilde{\mathbf{F}}(is) = \mathrm{Re}\left(\frac{5}{is - (-1 - 50i)} + \frac{0.2}{is - (-1 + 50i)} + 0.01\cos\left(\frac{9}{4}s\right) \cdot \mathbb{1}_{[-2\pi, 2\pi]}\right), \qquad s \in \mathbb{R},$$

using a trainable $\tilde{\mathbf{G}}$, where $9/4$ is chosen to guarantee the continuity of $\tilde{\mathbf{F}}$. We set the number of states to be one, i.e., $n = 1$. For illustration purposes, we fix $v = -1$ and $\zeta = 0$; therefore, we have

$$\tilde{\mathbf{G}}(is) = \mathrm{Re}\left(\frac{\xi}{is - (-1 - wi)}\right), \qquad s \in \mathbb{R},$$

where our only trainable parameters are $w$ and $\xi$. Our target function $\tilde{\mathbf{F}}$ contains two modes and some small noises between $-2\pi$ and $2\pi$, whereas $\tilde{\mathbf{G}}$ is unimodal with a trainable position and height (see Figure 3 (left)). We apply gradient flow on $w$ and $\xi$ with respect to the $L^2$-loss in the Fourier domain, in which case the functional derivative $(\partial/\partial\tilde{\mathbf{G}}(is))\mathcal{L}$ simply reduces to the residual:

$$\frac{\partial\mathcal{L}}{\partial\tilde{\mathbf{G}}(is)} = -2(\tilde{\mathbf{F}} - \tilde{\mathbf{G}})(is), \qquad \mathcal{L} = \|\tilde{\mathbf{F}}(is) - \tilde{\mathbf{G}}(is)\|_{L^2}.$$

In Figure 3 (middle), we show the training dynamics of $(w(\tau), \xi(\tau))$, initialized with different values $(w(0), \xi(0) = 3)$, where $\tau$ is the time index of the gradient flow. We make two remarkable observations that corroborate our discussion of the frequency bias during training:

1. Depending on the initialization of $w(0)$, it has two options of moving left or right. Since we fix $\zeta = 0$, by Theorem 1, a mode $(\hat{w}, \hat{\xi}) = (-50, 5)$ or $(50, 0.2)$ in the residual $\tilde{\mathbf{F}} - \tilde{\mathbf{G}}$ impacts the gradient $(\partial/\partial w)\mathcal{L}$ inverse-proportionally to the cube of the distance between the mode $\hat{w}$ and the current $w(\tau)$. Since $|24.5 - 50|^3/|24.5 - (-50)|^3 \approx 0.2/5$, we indeed observe that when $w(0) \leq 24.5$, it tends to move leftward, and rightward otherwise.

2. Although the magnitude of the noises in $[-2\pi, 2\pi]$ is only $5\%$ of the smaller mode at $\hat{w} = 50$ and $0.2\%$ of the larger mode at $\hat{w} = -50$, once $w(\tau)$ of the trainable LTI system $\tilde{\mathbf{G}}$ enters the noisy region, it gets stuck in a local minimum and never converges to one of the two modes of $\tilde{\mathbf{F}}$ (see Region II in Figure 3). This corroborates our discussion that the training dynamics of $w$ is sensitive to local information and it rarely learns the high frequencies when initialized in the low-frequency region.

## 5 TUNING THE FREQUENCY BIAS OF AN SSM

In section 3 and 4, we analyze the frequency bias of an SSM initialized by HiPPO and trained by a gradient-based algorithm. While we now have a theoretical understanding of the frequency bias, from a practical perspective, we want to be able to tune it. In this section, we design two strategies to enhance, reduce, counterbalance, or even reverse the bias of an SSM against the high frequencies. The two strategies are motivated by our discussion of the initialization (see section 3) and training (see section 4) of an SSM, respectively.

### 5.1 TUNING FREQUENCY BIAS BY SCALING THE INITIALIZATION

Since the initialization assigns an SSM some inborn frequency bias, a natural way to tune the frequency bias is to modify the initialization. Here, we introduce a hyperparameter $\alpha > 0$ as a simple way to scale the HiPPO initialization defined in Corollary 1:

$$a_j = -0.5 + i(-1)^j \lfloor j/2 \rfloor \alpha\pi, \qquad 1 \leq j \leq n. \tag{5}$$

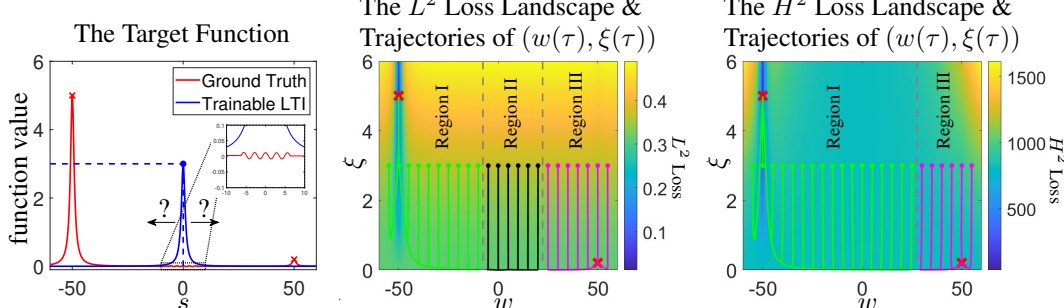

**Figure 3:** We train an LTI system to learn a noisy bimodal target transfer function. The convergence to a local minimum depends on the initial location of the pole. Left: the ground truth contains a large mode and a small mode, plus some small noises. We want to investigate which mode, if any, our trainable LTI system converges to. Middle: we train the LTI system with respect to the $L^2$-loss. We show the trajectories of $(w(\tau), \xi(\tau))$ given different initializations $(w(0), \xi(0) = 3)$. The two local minima corresponding to the two modes of $\tilde{\mathbf{F}}$ are shown in red crosses. The green trajectories (initialized in Region I) converge to the mode at $w = -50$, the magenta trajectories (initialized in Region III) converge to the mode at $w = 50$, and the black ones (initialized in Region II) converge to neither. Right: the experiment is repeated with the $H^2$-loss (see section 5.2).

Compared to the original HiPPO initialization, we scale the imaginary parts of the eigenvalues of $\mathbf{A}$ by a factor of $\alpha$. By making the modification, we lose the "polynomial projection" interpretation of HiPPO that was originally proposed as a way of explaining the success of the HiPPO initialization; yet, as shown in Orvieto et al. (2023); Yu et al. (2024a), this mechanism is no longer regarded as the key for a good initialization. By setting $\alpha < 1$, the eigenvalues $a_j$ are clustered around the origin, enhancing the bias against the high-frequency modes; conversely, choosing $\alpha > 1$ allows us to capture more variations in the high-frequency domain, reducing the frequency bias.

So far, our discussion in the paper is from a perspective of the continuous-time LTI systems acting on continuous time-series. For SSMs, however, the inputs come in a discrete sequence. Hence, we inevitably have to discretize our LTI systems. To study the scaling laws of $\alpha$, we assume in this work that an LTI system is discretized using the bilinear transform (Glover, 1984; Gu et al., 2022b) with a sampling interval $\Delta t > 0$. Other discretization choices can be similarly studied. Then, given an input sequence $\mathbf{u} \in \mathbb{R}^L$ of length $L$, the output $\mathbf{y}$ can be computed by discretizing eq. (2):

$$\mathbf{y} = \texttt{iFFT}(\texttt{FFT}(\mathbf{u}) \circ \tilde{\mathbf{G}}(\boldsymbol{s})), \qquad s_j = \frac{2}{\Delta t} \frac{\exp(i2\pi(j-1)/L) - 1}{\exp(i2\pi(j-1)/L) + 1}, \tag{6}$$

with the same transfer function $\tilde{\mathbf{G}}$ in eq. (3). Our goal is to propose general guidelines for an upper bound of $\alpha$. We leave most technical details to Appendix D; but we intuitively explain why $\alpha$ cannot be arbitrarily large for discrete inputs. Given a fixed sampling interval $\Delta t$, there is an upper bound for the frequency, called the Nyquist frequency, above which a signal cannot be "seen" by sampling, causing the so-called aliasing errors (Oppenheim, 1999; Condon & Ransom, 2016; Trefethen, 2019). As a straightforward example, one cannot distinguish between $\cos(5t)$ and $\cos(t)$ from their samples at $t = k\pi$, $k \in \mathbb{Z}$. Our next result tells us how to avoid aliasing by constraining the range of $\alpha$.

**Proposition 1.** Let $a = v + iw$ be given and define $\mathbf{G}(is) = 1/(is - a)$. Let $\boldsymbol{g} = \mathbf{G}(\boldsymbol{s})$ be the vector of length $L$, where $\boldsymbol{s}$ is defined in eq. (6). Then, there exist constants $C_1, C_2 > 0$ such that

$$C_1 \|\boldsymbol{g}\|_2 \le |w| \Delta t \le C_2 \|\boldsymbol{g}\|_2, \quad C_1 \|\boldsymbol{g}\|_\infty \le \frac{1}{1 + ||w| - 2(\Delta t)^{-1} \tan((1 - (L-1)/L)\pi/2)|} \le C_2 \|\boldsymbol{g}\|_\infty.$$

You may have noticed that in Proposition 1, we study the norm of the complex $\mathbf{G}$ instead of its real part restriction $\tilde{\mathbf{G}}$. The reason is that in an LTI system parameterized by complex numbers, we multiply $\mathbf{G}$ by a complex number $\xi + i\zeta$ and then extract its real part. Hence, both $\text{Re}(\mathbf{G})$ and $\text{Im}(\mathbf{G})$ are important. By noting that $\max_{1 \le j \le n} |w_j| \approx n\pi/2$ in our scaled initialization in eq. (5), Proposition 1 gives us two scaling laws of $\alpha$ that prevent $\|\boldsymbol{g}\|_2$ and $\|\boldsymbol{g}\|_\infty$ from vanishing, respectively. First, the 2-norm of $\boldsymbol{g}$ measures the average-case contribution of the partial fraction $1/(is - a)$ to the input-to-output mapping in eq. (6).

**Rule I:** *(Law of Non-vanishing Average Information) For a fixed task, as $n$ and $\Delta t$ vary, one should scale $\alpha = \mathcal{O}(1/(n\Delta t))$ to preserve the LTI system's impact on an average input.*

Next, the $\infty$-norm of $\boldsymbol{g}$ tells us the maximum extent of the system's action on any inputs. Therefore, if $\|\boldsymbol{g}\|_\infty$ is too small, then $1/(is - a)$ can be dropped without seriously affecting the system at all.

**Rule II:** *(Law of Nonzero Information) Regardless of the task, one should never take*

$$\alpha \gg 4 \tan\left((1-(L-1)/L)\,\pi/2\right)/n\pi\Delta t$$

*to avoid a partial fraction that does not contribute to the evaluation of the model.*

We reemphasize that our scaling laws provide *upper bounds* of $\alpha$. Of course, one can always choose $\alpha$ to be much smaller to capture the low frequencies better.

## 5.2 TUNING FREQUENCY BIAS BY A SOBOLEV FILTER

In section 5.1, we see that we can scale the HiPPO initialization to redefine the region in the Fourier domain that can be learned by an LTI system. Here, we introduce another way to tune the frequency bias: by applying a Sobolev-norm-based filter. The two strategies both tune the frequency bias, but by different means: scaling the initialization identifies a new set of frequencies that can be learned by the SSM, whereas the filter in this section introduces weights to different frequencies. Our method is rooted in the Sobolev norm, which extends a general $L^2$ norm. Imagine that we approximate a ground-truth transfer function $\tilde{\mathbf{F}}(is)$ using $\tilde{\mathbf{G}}(is)$. We can define the loss to be

$$\|\tilde{\mathbf{F}} - \tilde{\mathbf{G}}\|_{H^\beta}^2 := \int_{-\infty}^{\infty} (1 + |s|)^{2\beta} |\tilde{\mathbf{F}}(is) - \tilde{\mathbf{G}}(is)|^2 \, ds \tag{7}$$

for some hyperparameter $\beta \in \mathbb{R}$. The scaling factor $(1 + |s|)^{2\beta}$ naturally reweighs the Fourier domain. Note that you may have seen other forms of this factor — they all lead to the same norm up to norm-equivalency. When $\beta = 0$, eq. (7) reduces to the standard $L^2$ loss. The high frequencies become less important when $\beta < 0$ and more important when $\beta > 0$. Unfortunately, as discussed in section 4, there lacks a notion of the "ground-truth" $\tilde{\mathbf{F}}$ for every single LTI system within an SSM, making eq. (7) uncomputable. To address this issue, instead of using a Sobolev loss function, we apply a Sobolev-norm-based filter to the transfer function $\tilde{\mathbf{G}}$ to redefine the dynamical system:

$$\hat{\mathbf{y}}(s) = \tilde{\mathbf{G}}^{(\beta)}(is)\hat{\mathbf{u}}(s), \qquad \tilde{\mathbf{G}}^{(\beta)}(is) := (1 + |s|)^\beta \tilde{\mathbf{G}}(is). \tag{8}$$

This equation can be discretized using the same formula in eq. (6) by replacing $\tilde{\mathbf{G}}$ with $\tilde{\mathbf{G}}^{(\beta)}$.

Equation (8) can be alternatively viewed as applying the filter $(1 + |s|)^\beta$ to the FFT of the input $\mathbf{u}$, which clearly allows us to reweigh the frequency components. Surprisingly, there is even more beyond this intuition: applying the filter allows us to modify the training dynamics of $w_j$!

**Theorem 2.** Let $\mathcal{L}(\boldsymbol{\Theta})$ be a loss function and $\Gamma = (\mathbf{A}, \mathbf{B}, \mathbf{C}, \mathbf{D})$ be a diagonal LTI system in an SSM defined in section 3. For any $\beta \in \mathbb{R}$, we apply the filter in eq. (8) to $\Gamma$ and let $\tilde{\mathbf{G}}^{(\beta)}$ be the new transfer function. Suppose the functional derivative of $\mathcal{L}(\boldsymbol{\Theta})$ with respect to $\tilde{\mathbf{G}}^{(\beta)}(is)$ exists and is denoted by $(\partial/\partial\tilde{\mathbf{G}}^{(\beta)}(is))\mathcal{L}$. Then, if $|(\partial/\partial\tilde{\mathbf{G}}^{(\beta)}(is))\mathcal{L}| = \mathcal{O}(|s|^p)$ for some $p < 1 - \beta$, we have

$$\frac{\partial \mathcal{L}}{\partial w_j} = \int_{-\infty}^{\infty} \frac{\partial \mathcal{L}}{\partial \tilde{\mathbf{G}}^{(\beta)}(is)} \cdot K_j^{(\beta)}(s)ds, \quad K_j^{(\beta)}(s) := (1+|s|)^\beta \frac{\zeta_j((s-w_j)^2 - v_j^2) - 2\xi_j v_j(s-v_j)}{[v_j^2 + (s-w_j)^2]^2}, \tag{9}$$

for every $1 \le j \le n$. In particular, we have that $\left|K_j^{(\beta)}(s)\right| = \mathcal{O}\left(|\zeta_j s^{-2+\beta}| + |\xi_j s^{-3+\beta}|\right)$ as $|s| \to \infty$.

Compared to Theorem 1, the gradient of $\mathcal{L}$ with respect to $w_j$ now depends on the loss at frequency $s$ by a factor of $\mathcal{O}(|s|^{-2+\beta})$. Thus, the effect of our Sobolev-norm-based filter is not only a rescaling of the inputs in the frequency domain, but it also allows better learning the high frequencies:

*The higher the $\beta$ is, the more sensitive $w_j$ is to high-frequency losses. Hence, $w_j$ is no longer constrained by the "local-frequency" loss and will actively learn the high frequencies.*

The decay constraint that $|(\partial/\partial\tilde{\mathbf{G}}^{(\beta)}(is))\mathcal{L}| = \mathcal{O}(|s|^p)$ for some $p < 1 - \beta$ is needed to guarantee the convergence of the integral in eq. (9). When it is violated, the theoretical statement breaks, but

we could still implement the filter in practice, which is similar to Yu et al. (2023). In Figure 3 (right), we reproduce the illustrative example introduced in Section 4 using the Sobolev-norm-based filter in eq. (8) with $\beta = 2$. This is equivalent to training an ordinary LTI system with respect to the $H^2$-loss function defined in eq. (7). We find that in this case, the trajectories of $(w(\tau), \xi(\tau))$ always converge to one of the two modes in $\tilde{\mathbf{F}}$ regardless of the initialization, with more of them converging to the high-frequency global minimum on the left. This verifies our theory, because by setting $\beta = 2$, we amplify the contribution of the high-frequency residuals in the computation of $(\partial/\partial w)\mathcal{L}$, pushing a $y$ out of the noisy region between $-2\pi$ and $2\pi$. We leave more illustrative experiments to Appendix E, which show the effect of our tuning filter also when $\beta < 0$.

## 6 EXPERIMENTS AND DISCUSSIONS

**(I) SSMs as Denoising Sequential Autoencoders.** We now provide an example of how our two mechanisms allow us to tune frequency bias. In this example, we train an SSM to denoise an image in the CelebA dataset (Liu et al., 2015). We flatten an image into a sequence of pixels in the *row-major* order and feed it into an S4D model. We collect the corresponding output sequence and reshape it into an image. Similar to the setting of an autoencoder, our objective is to learn the identity map. To make the task non-trivial, we remove the skip connection $\mathbf{D}$ from the LTI systems. During inference, we add two different types of noises to the input images: horizontal or vertical stripes (see Figure 4). While the two types of noises may be visually similar to each other, since we flatten the images using the row-major order, the horizontal stripes turn into low-frequency noises while the vertical stripes become high-frequency ones (see Figure 9). In Figure 4, we show the outputs of the models trained with different values of $\alpha$ and $\beta$ as defined in section 5.1 and 5.2, respectively.

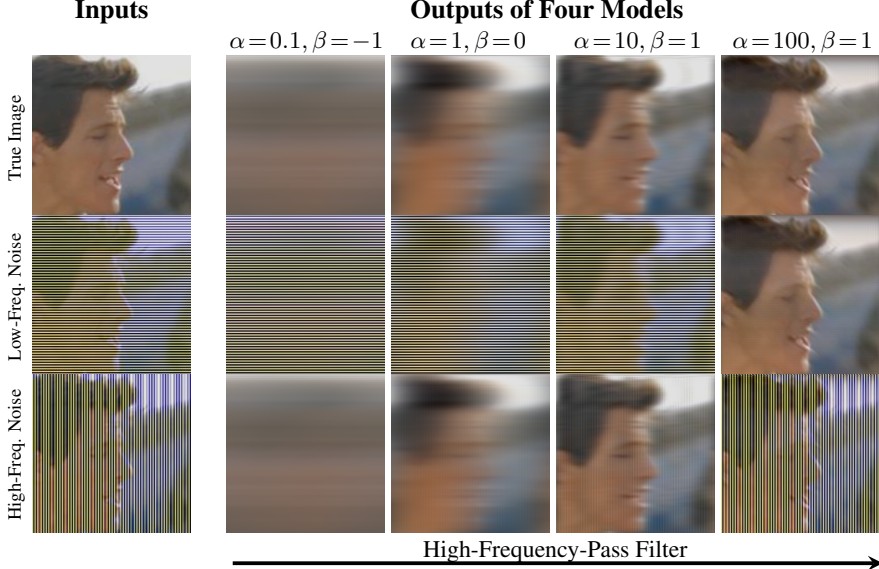

**Figure 4:** The outputs of image-denoising S4D models trained with different configurations.

From Figure 4, we can see that as $\alpha$ and $\beta$ increase, our model learns the high frequencies in the input better; consequently, the high-frequency noises get preserved in the outputs and the low-frequency noises are dampened. This corroborates our intuitions from section 5.1 and 5.2. We can further quantify the "pass rates" of the low and high-frequency noises. That is, we compute the percentage of the low and high-frequency noises that are preserved in the output. We show in Table 2 the ratio between the low-pass rate and the high-pass rate, which decreases as $\alpha$ and $\beta$ increase.

**(II) Tuning Frequency Bias in the Long-Range Arena.** The two tuning strategies section 5.1 and 5.2 are not only good when one needs to deal with a particular high or low frequency, but they also improve the performance of an SSM on general long-range tasks. In Table 3, we show that equipped with the two tuning strategies, our SSM achieves state-of-the-art performance on the Long-Range Arena (LRA) tasks (Tay et al., 2021).

**Table 2:** We compute the percentage of the low and high-frequency noises that are preserved in the output of a model trained with a pair of configurations $(\alpha, \beta)$. The table shows the ratio between the low-frequency pass rate and the high-frequency pass rate. The more bluish a cell is, the better our model learns the high frequencies. Circled in red is the S4D default.

| | | | | $\beta$ | | |
|---|---|---|---|---|---|---|
| | | $-1.0$ | $-0.5$ | $0.0$ | $0.5$ | $1.0$ |
| | 0.1 | 4.463e+07 | 2.409e+06 | 1.198e+05 | 4.613e+03 | 1.738e+02 |
| $\alpha$ | 1 | 4.912e+05 | 2.124e+05 | 1.758e+04 | 9.595e+02 | 5.730e+01 |
| | 10 | 9.654e+04 | 7.465e+03 | 6.073e+02 | 5.699e+01 | 6.394e+00 |
| | 100 | 3.243e+00 | 3.745e-02 | 3.801e-03 | 7.299e-05 | 5.963e-06 |

**Table 3:** Test accuracies in the Long-Range Arena of different variants of SSMs. The bold (resp. underlined) numbers indicate the best (resp. second best) performance on a task. An entry is left blank if no result is found. The row labeled "Ours" stands for the S4D model equipped with our two tuning strategies. Experiments were run with 5 random seeds and the medians and the standard deviations are reported. The S4 and S4D results are from the original papers (Gu et al., 2022b;a). The sizes of our models are the same or smaller than the corresponding S4D models.

| Model | ListOps | Text | Retrieval | Image | Pathfinder | Path-X | Avg. |
|---|---|---|---|---|---|---|---|
| DSS (Gupta et al., 2022) | 57.60 | 76.60 | 87.60 | 85.80 | 84.10 | 85.00 | 79.45 |
| S4++ (Qi et al., 2024) | 57.30 | 86.28 | 84.82 | 82.91 | 80.24 | - | - |
| Reg. S4D (Liu & Li, 2024a) | 61.48 | 88.19 | 91.25 | 88.12 | 94.93 | 95.63 | 86.60 |
| Spectral SSM (Agarwal et al., 2023) | 60.33 | 89.60 | 90.00 | - | 95.60 | 90.10 | - |
| Liquid S4 (Hasani et al., 2023) | **62.75** | 89.02 | 91.20 | 89.50 | 94.80 | 96.66 | 87.32 |
| S5 (Smith et al., 2023) | 62.15 | 89.31 | 91.40 | 88.00 | 95.33 | **98.58** | 87.46 |
| S4 (Gu et al., 2022b) | 59.60 | 86.82 | 90.90 | 88.65 | 94.20 | 96.35 | 86.09 |
| S4D (Gu et al., 2022a) | 60.47 | 86.18 | 89.46 | 88.19 | 93.06 | 91.95 | 84.89 |
| Ours | **62.75** ±0.78 | **89.76** ±0.22 | **92.45** ±0.16 | **90.89** ±0.35 | **95.89** ±0.13 | 97.84 ±0.21 | **88.26** |

**(III) Ablation Studies.** We perform ablation studies of our two tuning strategies by training a smaller S4D model to learn the grayscale sCIFAR-10 task. From Figure 5, we obtain better performance when we slightly increase $\alpha$ or decrease $\beta$. It might feel like a contradiction because increasing $\alpha$ helps to learn the high frequencies while decreasing $\beta$ downplays their role. It is not: $\alpha$ and $\beta$ control two different notions of the bias: scaling $\alpha$ affects "which frequencies we can learn;" scaling $\beta$ affects "how much we want to learn a certain frequency." They can be used collaboratively and interactively to attain the optimal extent of frequency bias that we need for a problem.

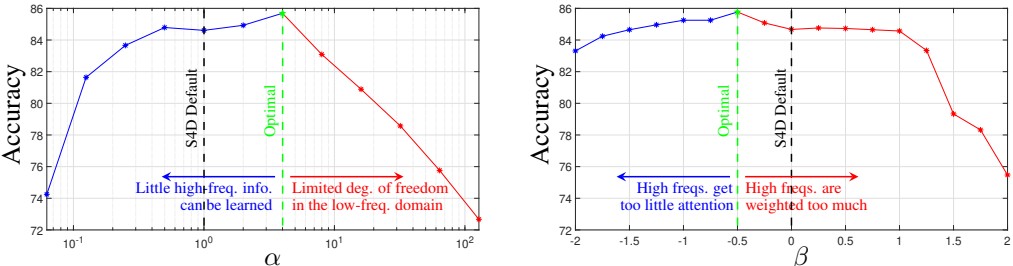

**Figure 5:** Two ablation studies of the tuning strategies proposed in this paper. We train an S4D model with varying parameters of $\alpha$ and $\beta$, respectively. On the left, we see that holding $\beta = 0$ (the default value), the model achieves its best performance when $\alpha = 4$; on the right, when we fix $\alpha = 1$ (the default value), the model performs the best when $\beta = -0.5$.

**(IV) More Experiments.** One can find more supplementary experiments in Appendix I on wave prediction (see Figure 1) and video generation.

## 7 CONCLUSION

We formulated the frequency bias of an SSM and showed its existence by analyzing both the initialization and training. We proposed two different tuning mechanisms based on scaling the initialization and on applying a Sobolev-norm-based filter to the transfer function. As a future direction, one could develop ways to analyze the spectral information of the inputs of a problem and use it to guide the selection of the hyperparameters in our tuning mechanisms.

ACKNOWLEDGMENTS

We would like to acknowledge the Laboratory Directed Research and Development Program of Lawrence Berkeley National Laboratory under U.S. Department of Energy Contract No. DE-AC02-05CH11231. In addition, SHL would like to acknowledge support from the Wallenberg Initiative on Networks and Quantum Information (WINQ) and the Swedish Research Council (VR/2021-03648). We would also like to thank the anonymous reviewers for their comments that helped improve this paper.

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

## A    MORE ON THE DEFINITION OF FREQUENCY BIAS

In this section, we provide an additional figure that explains the frequency bias of an SSM.

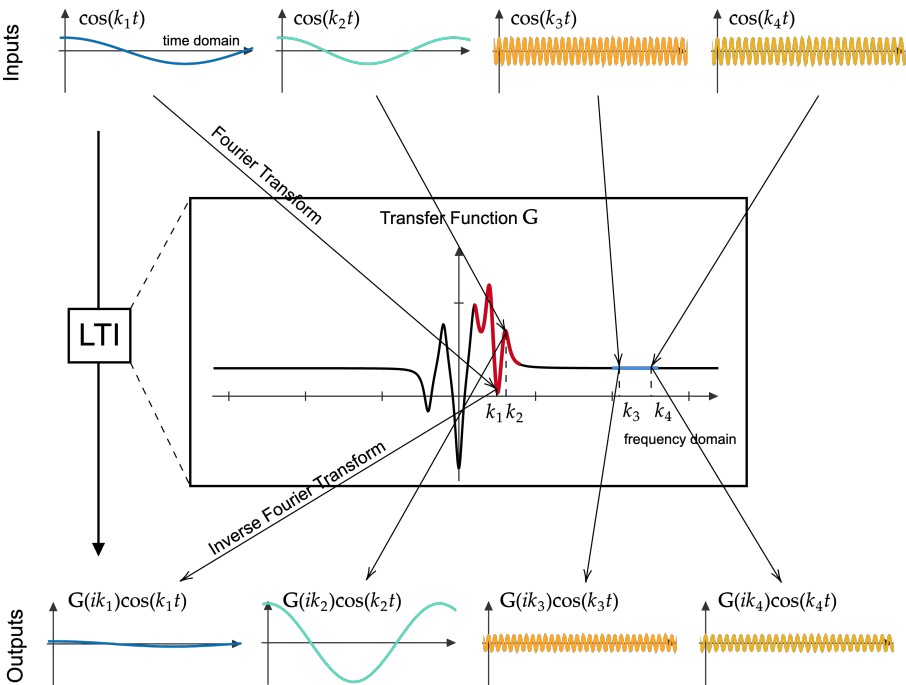

**Figure 6:** If the transfer function has a large total variation in the low-frequency region, then given two different low-frequency input signals, the LTI system sets very different pass rates for them. Conversely, when the transfer function has a small total variation in the high-frequency region, then given two different high-frequency input signals, the LTI system must set similar pass rates. The Fourier transform of a cosine wave involves both a positive and a negative frequency. We drop the negative frequency component for the cleanliness of the figure.

We also provide additional justifications for our definition of the frequency bias in section 2. As noted in Yu et al. (2024a), one reason why an LTI system in an SSM degenerates is that its transfer function is too flat. That is, if $\mathbf{G}$ can be well-approximated by a much lower-degree rational function, i.e.,

$$\inf_{\tilde{\mathbf{G}} \in \mathcal{R}^d} \sup_{s \in \mathbb{R}} |\mathbf{G}(is) - \tilde{\mathbf{G}}(is)|$$

is small for some $d \ll n$, where $\mathcal{R}^d$ is the space of rationals of degree $\leq d$, then the large LTI system can be well-approximated by a much smaller one (i.e., given the same input $\mathbf{u}$, the outputs of the two systems are close to each other). In other words, our LTI system is not as expressive as its size may have suggested. If we now restrict our attention to only a part of the frequency domain, then the same reasoning applies: if the transfer function $\mathbf{G}$ is flat on $[a, b]$, then that means

$$\inf_{\tilde{\mathbf{G}} \in \mathcal{R}^d} \sup_{s \in [a,b]} |\mathbf{G}(is) - \tilde{\mathbf{G}}(is)|$$

is small for some $d \ll n$. Therefore, there exists a much smaller LTI system $\tilde{\Gamma}$ and its actions on the Fourier modes in $[a, b]$ are very similar to those of the original system. Hence, this essentially means that our LTI system is unable to capture "complex patterns" in the frequency interval $[a, b]$ because all it does in $[a, b]$ can also be done by a much smaller system.

## B    PROOFS

In this section, we provide the proofs of all theoretical statements in the manuscript.

First, we prove the statements about the initialization of the LTI systems. The total variation can be bounded straightforwardly using the decay of the transfer functions.

*Proof of Lemma 1.* Since $\tilde{\mathbf{G}}$ is the real part of $\mathbf{G}$, its total variation is always no larger than the total variation of $\mathbf{G}$. Hence, we have

$$V_{-\infty}^{-B}(\tilde{\mathbf{G}}) \leq V_{-\infty}^{-B}(\mathbf{G}) \leq \sum_{j=1}^{n} \left| \frac{c_j}{-iB - a_j} \right| \leq \sum_{j=1}^{n} \left| \frac{c_j}{B + w_j} \right|.$$

The other bound is similarly obtained. $\qquad\square$

*Proof of Corollary 1.* By Lemma 1 and the Hölder's inequality, we have

$$V_{-\infty}^{-B}(\tilde{\mathbf{G}}) \leq \sum_{j=1}^{n} \frac{|c_j|}{|w_j + B|} \leq \|\mathbf{B} \circ \mathbf{C}^\top\|_2 \|\mathbf{w}\|_2,$$

where $\mathbf{w}_j = 1/(w_j + B)$. Since $\xi_j, \zeta_j \sim \mathcal{N}(0,1)$ i.i.d., we have that $\|\mathbf{B} \circ \mathbf{C}^\top\|_2^2$ follows the $\chi^2$-distribution with degree $2n$. By Laurent & Massart (2000), we have with probability at least $1 - \delta$ that

$$\|\mathbf{B} \circ \mathbf{C}^\top\|_2^2 \leq 2n + \sqrt{2n \ln(1/\delta)} + 2\ln(1/\delta) \leq 2(\sqrt{n} + \sqrt{\ln(1/\delta)})^2.$$

By the definition of $B$, we have that

$$\|\mathbf{w}\|_2^2 \leq \frac{n}{(B - n/2)^2}.$$

The result for $V_{-\infty}^{-B}(\tilde{\mathbf{G}})$ follows from the last two inequalities. The bound on $V_B^\infty(\tilde{\mathbf{G}})$ can be derived similarly. $\qquad\square$

The initialization in Corollary 1 is called the HiPPO-Lin initialization. Using the same idea, we can immediately prove a result for the HiPPO-LegS initialization.

**Corollary 2.** Assume that the diagonal entries $a_j$ are initialized using HiPPO-LegS (Gu et al., 2022a) and $\xi_j, \zeta_j \sim \mathcal{N}(0,1)$ i.i.d., where $\mathcal{N}(0,1)$ is the standard normal distribution. Then, given $B > 0$ and $\delta > 0$, we have

$$V_{-\infty}^{-B}(\tilde{\mathbf{G}}), V_B^\infty(\tilde{\mathbf{G}}) \leq \frac{\sqrt{2n}(\sqrt{n} + \sqrt{\ln(1/\delta)})}{B} \qquad \text{with probability} \geq 1 - \delta.$$

*Proof.* The proof is done by noting that every $a_j$ is real-valued and mimicing the proof of Corollary 1. $\qquad\square$

We skip the proof of Proposition 1 and defer it to Appendix D when we present a detailed derivation of the scaling laws. We next prove the statement about the training dynamics of the imaginary parts of $\text{diag}(\mathbf{A})$.

*Proof of Theorem 1.* Fix some $1 \leq j \leq n$, we first view the transfer function $\tilde{\mathbf{G}}(s, w_j)$ as a function of two variables. We compute the derivative of $\tilde{\mathbf{G}}(s, w_j)$ with respect to $w_j$:

$$\frac{\partial \tilde{\mathbf{G}}(s, w_j)}{\partial w_j} = \frac{\partial}{\partial w_j} \left( \sum_{j=1}^{n} \frac{\zeta_j(s - w_j) - \xi_j v_j}{v_j^2 + (s - w_j)^2} \right) + \mathbf{D} = \frac{\partial}{\partial w_j} \left( \frac{\zeta_j(s - w_j) - \xi_j v_j}{v_j^2 + (s - w_j)^2} \right)$$

$$= \frac{(v_j^2 + (s - w_j)^2) \cdot (\partial/\partial w_j)(\zeta_j(s - w_j) - \xi_j v_j)}{(v_j^2 + (s - w_j)^2)^2}$$

$$- \frac{(\zeta_j(s - w_j) - \xi_j v_j) \cdot (\partial/\partial w_j)(v_j^2 + (s - w_j)^2)}{(v_j^2 + (s - w_j)^2)^2}$$

$$= \frac{-(v_j^2 + (s - w_j)^2)\zeta_j}{(v_j^2 + (s - w_j)^2)^2} + \frac{2(\zeta_j(s - w_j) - \xi_j v_j)(s - w_j)}{(v_j^2 + (s - w_j)^2)^2}$$

$$= \frac{\zeta_j(-v_j^2 - (s - w_j)^2 + 2(s - w_j)^2) - \xi_j(2v_j(s - w_j))}{(v_j^2 + (s - w_j)^2)^2} = K_j(s).$$

Since we assume that $|(\partial/\partial\tilde{\mathbf{G}}(is))\mathcal{L}| = \mathcal{O}(|s|^p)$ for some $p < 1$, the integral in eq. (4) converges. By Parr & Yang (1994, Appendix A), eq. (4) holds. $\qquad\square$

The statement about the training dynamics of $w_j$ given a Sobolev filter follows immediately from Theorem 1.

*Proof of Theorem 2.* Since we assume that $|(\partial/\partial\tilde{\mathbf{G}}(is))\mathcal{L}| = \mathcal{O}(|s|^p)$ for some $p < 1 - \beta$, the integral in eq. (9) converges. The result follows by noting that

$$\frac{\partial\tilde{\mathbf{G}}^{(\beta)}(s, w_j)}{\partial w_j} = (1 + |s|)^\beta \frac{\partial\tilde{\mathbf{G}}(s, w_j)}{\partial w_j} = (1 + |s|)^\beta K_j(s) = K_j^{(\beta)}(s)$$

and applying the equation in Parr & Yang (1994, Appendix A). $\qquad\square$

## C  FUNCTIONAL DERIVATIVES

In this section, we briefly introduce the notion of functional derivatives (see Appendix A.1 in (Lim, 2021) for a more technical overview). To make our discussion concrete, we do it in the context of Theorem 1. Consider the transfer function $\tilde{\mathbf{G}}(is)$ defined in eq. (3). It depends on the model parameters $v_j, w_j, \xi_j$, and $\zeta_j$. In this section, we separate out a single $w_j$ for a fixed $1 \le j \le n$, leaving the remaining parameters unchanged. Then, for every $w \in \mathbb{R}$, we can define $f^{(w)}(s)$ to be the transfer function $\tilde{\mathbf{G}}(is)$ when $w_j = w$. Under this setting, the set of all possible transfer functions indexed by $w$, i.e., $\mathcal{F} = \{f^{(w)}|w \in \mathbb{R}\}$ is a subset of a Banach space, say $L^2(\mathbb{R})$. To avoid potential confusions, we shall remark that $f^{(w)}$ is not linear in its index $w$, i.e., $f^{(w_1+w_2)} \ne f^{(w_1)} + f^{(w_2)}$ in general, neither is $\mathcal{F}$ a subspace of $L^2(\mathbb{R})$. This does not impact our following discussion.

Now, consider the loss function $\mathcal{L}$. Given a choice of $w_j = w$ and a corresponding transfer function $f^{(w)}$, the loss function maps $f^{(w)}$ to a real number that corresponds on the current loss. Hence, $\mathcal{L}$ can be viewed as a (not necessarily linear) functional of $f^{(w)}$. We would like to ask: how does $\mathcal{L}$ respond to a small change of $f^{(w)}(s)$ at some $s \in \mathbb{R}$? Ideally, this can be measured as

$$\lim_{\epsilon\to 0} \frac{\mathcal{L}(f^{(w)} + \epsilon\delta_s) - f^{(w)}}{\epsilon}, \tag{10}$$

where $\delta_s$ is the Dirac delta function at $s$. However, the loss function $\mathcal{L}$ is not defined for distributions, making eq. (10) not directly well-defined. To fix this issue, we have to go through the functional derivatives. The idea, as usual in functional analysis, is to pass the difficulty of handling a distribution to smooth functions that approximate it. If there exists a function $(\partial/\partial f^{(w)})\mathcal{L}$ such that the equation

$$\int_{-\infty}^{\infty} \frac{\partial\mathcal{L}}{\partial f^{(w)}}(s)\phi(s)\ ds = \lim_{\epsilon\to 0} \frac{\mathcal{L}(f^{(w)} + \epsilon\phi) - \mathcal{L}(f^{(w)})}{\epsilon}$$

holds for all smooth $C_0^\infty$ functions $\phi$ that are infinitely differentiable and vanish at infinity, then $(\partial/\partial f^{(w)})\mathcal{L}$ is defined to be the functional derivative of $\mathcal{L}$ at $f^{(w)}$. Taking $\{\phi_j\}_{j=1}^{\infty}$ to be an approximate identity centered at $s$, we recover eq. (10) using $(\partial/\partial f^{(w)})\mathcal{L}(s)$.

One nice thing about the functional derivatives is that they allow us to write down a continuous analog of the chain rule, which is the meat of Theorem 1. To get some intuition, let us first consider a function $\tilde{\mathcal{L}}(w) = \tilde{\mathcal{L}}(f_1(w), \ldots, f_k(w))$ that depends on $w$ via $k$ intermediate variables $f_1, \ldots, f_k$. Assuming sufficient smoothness conditions, the derivative of $\tilde{\mathcal{L}}$ with respect to $w$ can be calculated using the standard chain rule:

$$\frac{\partial\tilde{\mathcal{L}}}{\partial w} = \sum_{j=1}^{k} \frac{\partial\tilde{\mathcal{L}}}{\partial f_j}\frac{\partial f_j}{\partial w} = \begin{bmatrix} \frac{\partial\tilde{\mathcal{L}}}{\partial f_1} & \cdots & \frac{\partial\tilde{\mathcal{L}}}{\partial f_k} \end{bmatrix} \begin{bmatrix} \frac{\partial f_1}{\partial w} \\ \vdots \\ \frac{\partial f_k}{\partial w} \end{bmatrix}. \tag{11}$$

The only difference in the case of $\mathcal{L}$ is that instead of depending on $k$ discrete intermediate variables $f_1, \ldots, f_k$, our $\mathcal{L}$ depends on a continuous family of intermediate variables $f^{(w)}(s)$ indexed by $s \in \mathbb{R}$. In this case, one would naturally expect that in eq. (11), the sum becomes an integral, or equivalently, the row and the column vectors become the row and the column functions. This is indeed the case given our functional derivative:

$$\frac{\partial \mathcal{L}}{\partial w} = \int_{-\infty}^{\infty} \frac{\partial \mathcal{L}}{\partial f^{(w)}}(s) \frac{\partial f^{(w)}(s)}{\partial w} \, ds.$$

This formula can be found in Parr & Yang (1994, (A.24)) and Greiner & Reinhardt (2013, sect. 2.3).

## D    SCALING LAWS OF THE INITIALIZATION

In this section, we expand our discussions in section 5.1 and give the proof of Proposition 1. Throughout this section, we assume that we use the bilinear transform to discretize our continuous-time LTI system. The length of our sequence is $L$ and the sampling interval is $\Delta t$. The bilinear transform is essentially a Mobius transform between the closed left half-plane of the $s$-domain and the closed unit disk in the $z$-domain. Hence, it gives us two ways to study this filter — by either transplanting the transfer function $\tilde{\mathbf{G}}$ onto the unit circle and analyzing in the discrete domain or by transplanting the FFT nodes from the $z$-domain to the imaginary axis in the $s$-domain. The two ways are equivalent, but we choose the second method for simplicity.

The output of an LTI system can be computed by

$$\mathbf{y} = \texttt{iFFT}(\texttt{FFT}(\mathbf{u}) \circ \overline{\mathbf{G}}(\boldsymbol{\omega})),$$

where $\overline{\mathbf{G}}$ is the transfer function of the discrete system and where

$$\boldsymbol{\omega} = \begin{bmatrix} \exp\left(2\pi i \frac{0}{L}\right) & \cdots & \exp\left(2\pi i \frac{L-1}{L}\right) \end{bmatrix}^{\top}$$

is the length-$L$ vector consisting of $L$th roots of unity. We do not have direct access to $\overline{\mathbf{G}}$, but we do know $\tilde{\mathbf{G}}$, its continuous analog, in the partial fractions format. They are related by the following equation:

$$\overline{\mathbf{G}}(z) = \tilde{\mathbf{G}}(s), \qquad s = \frac{2}{\Delta t} \frac{z-1}{z+1}.$$

In that case, the vector $\overline{G}(\boldsymbol{\omega})$ can be equivalently written as

$$\overline{\mathbf{G}}(\boldsymbol{\omega}_j) = \tilde{\mathbf{G}}\left(\frac{2}{\Delta t} \frac{\exp\left(2\pi i \frac{j}{L}\right) - 1}{\exp\left(2\pi i \frac{j}{L}\right) + 1}\right) = \tilde{\mathbf{G}}\left(i \frac{2}{\Delta t} \tan\left(\pi \frac{j}{L}\right)\right).$$

This is how we obtained eq. (6). The locations of the new samplers on the imaginary axis are shown in Figure 7, with $L = 101$ and $\Delta t = 0.01$.

Note that the right figure (the $s$-domain) is on a logarithmic scale and only the upper half-plane is shown due to the scale. We also choose $L$ odd; when $L$ is even, a pole is placed "at infinity" in the $s$-domain, at which any partial fraction vanishes. Why do we go through all the pains to study this bilinear transformation? The reason is that it gives us a guideline for scaling the poles. For instance, for $L = 101$ and $\Delta t = 0.01$, if a pole has a much larger imaginary part than $10^4$, then the discrete sequence will hardly see the effect of this partial fraction even though the underlying continuous system will. This corresponds to the intuition behind the aliasing error that we discussed in the main text.

As in Proposition 1, it suffices to study a single partial fraction instead of all. Hence, instead of studying the entire transfer function together, we focus on one component of it:

$$\mathbf{G}(is) = \frac{1}{is - a}, \qquad \text{Re}(s) = 0, \quad \text{Re}(a) < 0.$$

This is a partial fraction in the $s$-domain. For a fixed $L$ and $\Delta t$, this partial fraction corresponds to a bounded linear operator $\mathcal{G} : \ell^2([L]) \to \ell^2([L])$ that maps an input sequence to an output sequence, where $[L] = \{1, \ldots, L\}$ is the set of the first $L$ natural numbers. We consider the norm of this

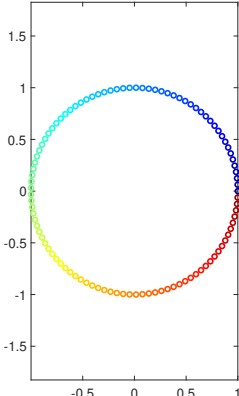 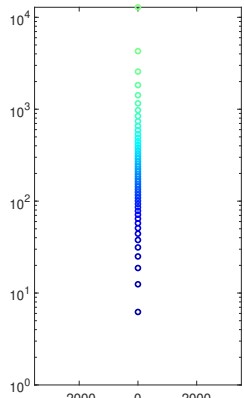

**Figure 7:** The poles of the FFT samplers in the $z$-domain and the samplers in the $s$-domain for $L = 101$ and $\Delta t = 0.01$. Due to the essential difficulty of plotting the entire real line on a logarithmic scale, we only show the semilog plot of the upper half-plane of the $s$-domain (right). Hence, on the right figure, we omit the samplers corresponding to the lower arc of the unit circle on the left.

operator, where we will find that as $|\text{Im}(s)| \to \infty$, the norm of the operator vanishes. The rate of vanishing will guide us in selecting an appropriate range for the pole. So, how can we tell the norm of this operator? By definition, the norm of $\mathcal{G}$ is defined by

$$\|\mathcal{G}\|_{\ell^2 \to \ell^2} = \sup_{\|\mathbf{u}\|_{\ell^2}=1} \|\mathbf{y}\|_{\ell^2} = \sup_{\|\hat{\mathbf{u}}\|_{\ell^2}=1} \|\hat{\mathbf{y}}\|_{\ell^2},$$

where $\mathbf{y}$ is the output of the operator $\mathcal{G}$ given input $\mathbf{u}$, i.e., $\mathbf{y} = \mathcal{G}\mathbf{u}$, and the second step follows from the Parseval's identity. By Hölder's inequality, we further have

$$\|\mathcal{G}\|_{\ell^2 \to \ell^2} = \sup_{\|\hat{\mathbf{u}}\|_{\ell^2}=1} \|\hat{\mathbf{u}} \circ \overline{\mathbf{G}}(\boldsymbol{\omega})\|_{\ell^2} \leq \|\boldsymbol{g}\|_{\ell^\infty},$$

where $\overline{\mathbf{G}}(\boldsymbol{\omega}) = \boldsymbol{g}$ is the sample vector of the bilinearly transformed transfer function of $\mathbf{G}$ in the $z$-domain, i.e.,

$$\boldsymbol{g} = \overline{\mathbf{G}}(\boldsymbol{\omega}), \qquad \overline{\mathbf{G}}(\boldsymbol{\omega})_j = \overline{\mathbf{G}}(\omega_j) = \mathbf{G}\left(i\frac{2}{\Delta t}\tan\left(\pi\frac{j}{L}\right)\right) = \frac{1}{i2\tan\left(\pi j/L\right)/\Delta t - a}.$$

Hence, we have

$$|\boldsymbol{g}_j|^2 = \frac{1}{\text{Re}(a)^2 + (\text{Im}(a) - 2\tan\left(\pi j/L\right)/\Delta t)^2}.$$

When $\text{Im}(a) > 2\tan\left(\pi j/L\right)/\Delta t$ for all $j$, $|\boldsymbol{g}_j|^2$ is maximized when $j = \lfloor L/2 \rfloor - 1$, in which case we have

$$\|\mathcal{G}\|_{\ell^2 \to \ell^2}^2 \leq \frac{1}{\text{Re}(a)^2 + (\text{Im}(a) - (2/\Delta t)\tan\left((\pi/2)(1 - (L-1)/L)\right))^2}. \tag{12}$$

This gives us the second rule (Law of Zero Information) when scaling the diagonal of $\mathbf{A}$. This is a worst-case analysis, where we essentially assume that the Fourier coefficient of $\mathbf{u}$ is one-hot at the highest frequency. In practice, of course, this assumption is a bit unrealistic; in fact, the Fourier coefficients usually decay as the frequency gets higher. Therefore, we should derive another rule for the average-case scenario. We consider the operator $\hat{\mathcal{G}}$ that maps the Fourier coefficients of the inputs to those of the outputs, then the norm $\|\hat{\mathcal{G}}\|_{\ell^\infty \to \ell^2}$ is a good average-case estimate, because

$$\arg\max_{\|\hat{\mathbf{u}}\|_{\ell^\infty}=1} \|\hat{\mathcal{G}}\hat{\mathbf{u}}\|_{\ell^2}$$

is necessarily at a vertex of the simplex defined by $\|\hat{\mathbf{u}}\|_{\ell^\infty} \leq 1$ [1]. That is, $\hat{\mathbf{u}}_j = \pm 1$ for all $1 \leq j \leq L$. Now, using the Hölder's inequality again, we have

$$\|\hat{\mathcal{G}}\|_{\ell^\infty \to \ell^2} = \sup_{\|\hat{\mathbf{u}}\|_{\ell^\infty}=1} \|\hat{\mathbf{u}} \circ \boldsymbol{g}\|_{\ell^2} \leq \|\boldsymbol{g}\|_{\ell^2}.$$

---

[1] Note that we can use max instead of sup because the domain $\{\|\hat{\mathbf{u}}\|_{\ell^\infty} = 1\}$ is compact.

Hence, instead of studying the $\ell^\infty$-norm of $\boldsymbol{g}$, we consider the $\ell^2$-norm for the average-case estimate. The precise computation of the $\ell^2$-norm can be hard, but let us write out the full expression:

$$\|\boldsymbol{g}\|_{\ell^2}^2 = \sum_{j=1}^{L} |\boldsymbol{g}|^2 \leq \sum_{j=1}^{L} \frac{1}{\mathrm{Re}(a)^2 + \left(\mathrm{Im}(a) - (2/\Delta t)\tan(\pi j/L)\right)^2}.$$

Given the imaginary part of $a > 0$, we grab all Fourier nodes on the $j\omega$ axis that are below $\mathrm{Im}(a)/2$ and lower-bound them; we also grab all above $\mathrm{Im}(a)/2$ and assume that they collapse to $\mathrm{Im}(a)$. This gives us an estimate of the $\ell^2$ norm:

$$
\begin{aligned}
\|\boldsymbol{g}\|_{\ell^2}^2 &\leq \left( \frac{|\{j|(2/\Delta t)\tan(\pi j/L) \leq \mathrm{Im}(a)/2\}|}{\mathrm{Re}(a)^2 + \mathrm{Im}(a)^2/4} + \frac{|\{j|(2/\Delta t)\tan(\pi j/L) > \mathrm{Im}(a)/2\}|}{\mathrm{Re}(a)^2} \right) \\
&\leq \left( \frac{L(2\arctan(\mathrm{Im}(a)\Delta t/4)/\pi)+1}{\mathrm{Re}(a)^2 + \mathrm{Im}(a)^2/4} + \frac{L(1 - 2\arctan(\mathrm{Im}(a)\Delta t/4)/\pi)+1}{\mathrm{Re}(a)^2} \right) \\
&= L\left( \underbrace{\frac{(2\arctan(\mathrm{Im}(a)\Delta t/4)/\pi)+1/L}{\mathrm{Re}(a)^2 + \mathrm{Im}(a)^2/4}}_{N_1} + \underbrace{\frac{(1 - 2\arctan(\mathrm{Im}(a)\Delta t/4)/\pi)+1/L}{\mathrm{Re}(a)^2}}_{N_2} \right)
\end{aligned}
\tag{13}
$$

*Proof of Proposition 1.* Given eq. (12) and (13). Proposition 1 follows immediately. □

Let us take a closer look at this expression. Ideally, the $\ell^2$-norm of $\boldsymbol{g}$ should be independent of $L$ and $\Delta t$; that is, as $L \to \infty$ and $\Delta t \to 0$, we do not want $\|\boldsymbol{g}\|_{\ell^2}^2/L$ to diminish. First, we note that $N_1$ and $N_2$ are independent of $L$ as $L \to \infty$. As $\Delta t \to 0$, $N_1$ inevitably vanish, regardless of the location of $\mathrm{Im}(a)$. In order to maintain $N_2$ a constant, we would need $\mathrm{Im}(a)\Delta t/4$ to not blow up. This gives us the first rule (Law of Zero Information) for scaling the poles. We can further work out some constants in $\mathcal{O}$ to be used in practice. For example, to guarantee that $\mathrm{Im}(a)$ is smaller than the top $5\%$ Fourier nodes, we would need that

$$\frac{2}{\pi}\arctan\left(\frac{\mathrm{Im}(a)\Delta t}{4}\right) \leq 0.95 \Rightarrow \mathrm{Im}(a) \leq \frac{50.82}{\Delta t}.$$

In particular, eq. (12) and eq. (13) together give us the proof of the lower bounds in Proposition 1. The upper bounds are proved by noting all all derivations in this section are asymptotically tight.

# E   MORE NUMERICAL EXPERIMENTS ON THE ILLUSTRATIVE EXAMPLE

In section 4 and 5.2, we see that using a Sobolev-norm-based filter with $\beta > 0$, one is able to escape from the local minima caused by small local noises. In this section, we present a similar set of experiments to show the effect of our filter, even when $\beta < 0$. We choose our new objective function to be

$$\tilde{\mathbf{F}}(is) = \mathrm{Re}\left( \frac{5}{is - (-1 - 75i)} + \frac{0.2}{is - (-1 + 25i)} \right), \qquad s \in \mathbb{R}.$$

Compared to the objective in section 4, we see two differences. First, we remove the sinusoidal noises around the origin. Second, we shift the locations of the two modes in the target: instead of locating at $s = -50$ and $s = 50$, we shift them to $s = -75$ and $s = 25$, respectively. This allows us to have a large high-frequency mode and a small low-frequency mode in the ground truth.

We show the results in Figure 8 when we train an LTI system using a Sobolev-norm-based filter with different values of $\beta$. Note that when $\beta = 0$, the picture only differs from Figure 1 (middle) in the frequency labels because in that case, the gradient $(\partial/\partial w)\mathcal{L}$ only cares about the relative difference $w - s$ but not the absolute values of $s$. From Figure 8, we see that as $\beta$ increases, more trajectories converge to the local minimum near $(\xi = 5, w = -75)$. The reason is that a larger $\beta$ favors a higher frequency (see Theorem 2).

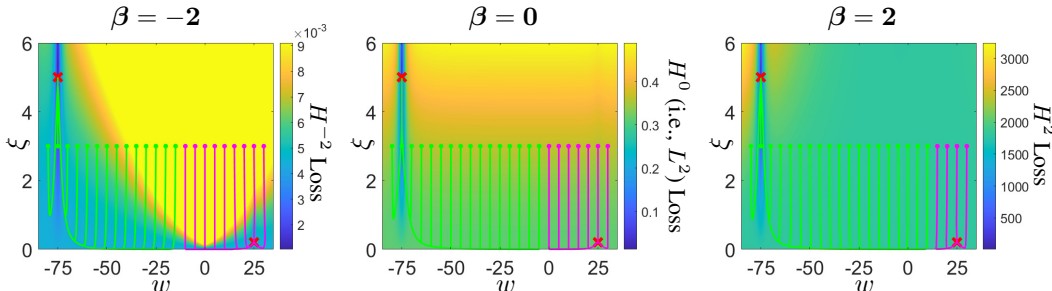

**Figure 8:** An SSM is trained with a filter based on the Sobolev-norm (see section 5.2). The plots are read in the same way as those in Figure 1. The transfer function converges to one of the two local minima. As $\beta$ ranges from $-2$ to $2$, the transfer function becomes more sensitive to the high-frequency global information rather than the local information. Hence, the edge between the two different convergences shifts rightward.

## F  IMPLEMENTATION OF THE SOBOLEV-NORM-BASED FILTER

There are two popular ways to evaluate an LTI system: via a convolutional kernel (Gu et al., 2022b) or via parallel scans (Smith et al., 2023). The first approach is directly based on the frequency domain computation in eq. (8); thus, the implementation is trivial. The parallel scan algorithm does not operate in the frequency domain, making the computation of eq. (8) less trivial. However, by noting that

$$\hat{\mathbf{y}}(s) = \tilde{\mathbf{G}}^{(\beta)}(is)\hat{\mathbf{u}}(s) = \tilde{\mathbf{G}}(is)\left[(1+|s|)^\beta \hat{\mathbf{u}}(s)\right],$$

one can imagine that we apply the standard parallel scan algorithm to the transformed inputs $\left[(1+|s|)^\beta \hat{\mathbf{u}}(s)\right]^\vee$. Hence, all we need to do is to preprocess the inputs by applying the filter $(1+|s|)^\beta$ in the Fourier domain and then use the standard parallel scan algorithm.

## G  CHOICE OF THE SCALING FACTOR

In this section, we briefly provide a practical guideline for how $\alpha$ can be chosen. We will advocate two ways to select an $\alpha_{\max}$ as an upper bound of $\alpha$. Then, given $\alpha_{\max}$, one needs to tune a hyper-parameter $\alpha \in (0, \alpha_{\max})$ that gives a satisfactory amount of frequency bias. We do not recommend tuning $\alpha$ together with other hyperparameters. Instead, we suggest starting with $\alpha = \alpha_{\max}$, and once all the rest of the hyperparameters are chosen, one can tune $\alpha$ using the bisection method, which only requires $\mathcal{O}(\log \alpha_{\max})$ many trials. The reason why the bisection method works is that we believe in a unique local minimum of the loss as $\alpha$ changes (see Figure 5).

To select an $\alpha_{\max}$, we can either be informed by the input data or not. If we choose not to study the input data, then we propose to guarantee that $\text{Im}(a)$ is smaller than the top $10\%$ Fourier nodes at which we sample the transfer function $\mathbf{G}$ (see Appendix D). That is,

$$\frac{2}{\pi}\arctan\left(\frac{\text{Im}(a)\Delta t}{4}\right) \le 0.9 \Rightarrow \text{Im}(a) \le \frac{25.26}{\Delta t} \Rightarrow \alpha_{\max} = \frac{50.52}{\pi n \Delta t}.$$

The other way is to select $\alpha_{\max}$ based on the input data. That is, one can plot the densities of the FFTs of all input training data and identify an edge $s_{\max}$ for which $[-s_{\max}, s_{\max}]$ contains most densities. Since the Fourier domain is one-dimensional, one can identify $s_{\max}$ by simply eyeballing. Then, we can select $\alpha_{\max}$ so that the information in $[-s_{\max}, s_{\max}]$ can be learned. That is,

$$\text{Im}(a) \le \frac{s_{\max}}{L\Delta t} \Rightarrow \alpha_{\max} = \frac{s_{\max}}{\pi n L \Delta t},$$

where $L$ is the length of the sequence.

## H  DETAILS OF THE EXPERIMENTS

### H.1  DENOISING SEQUENTIAL AUTOENCODER

For every image in the CelebA dataset, we reshaped it to have a resolution of $1024 \times 256$ pixels to allow for higher-frequency noises. We trained a single-layer S4D model with $n = 128$ and `d_model` $= 3$. We dropped the skip connection $\mathbf{D}$ from the model. The model was trained using the MSE loss. That is, for every predicted sequence of pixels, we compared the model against the true image and computed the 2-norm of the difference vector.

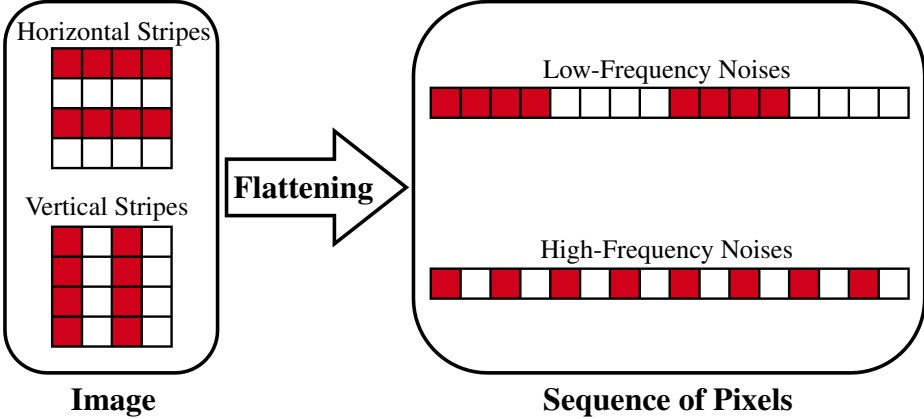

**Figure 9:** When a noisy image is flattened into pixels using the row-major order, the noises induced by horizontal stripes become low-frequency noises while those induced by vertical stripes become high frequencies.

We trained the model on the original images, i.e., those without any noises. When we inferred from the model, we added noises to the inputs, introducing about 10 cycles of horizontal or vertical stripes, respectively. Our noises were large, almost shielding the underlying images. When the value of a pixel was out of range, then we ignored such as issue during training; we clipped its value to the appropriate range when rendering the image in Figure 4.

To obtain the numbers in Table 2, we computed with our trained models, where we set the inputs to be pure horizontal or vertical noises with no underlying images. Then, we evaluated the size of the output image and took the ratio of the outputs over the inputs. We call this value the "pass rate" of a particular noise. Table 2 shows the ratio between the pass rate of the low-frequency noises over the high-frequency ones. Our model did not have a nonlinear activation function, which made the model linear. Hence, it does not matter what the magnitude of the inputs was.

### H.2  LONG-RANGE ARENA

In this section, we present the hyperparameters of our models trained on the Long-Range Arena tasks. Our model architecture and hyperparameters are almost identical to those of the S4D models reported in Gu et al. (2022a), with only two exceptions: for the `ListOps` experiment, we set $n = 2$ instead of $n = 64$, which aligns with Smith et al. (2023) instead; for the `PathX` experiment, we set `d_model` $= 128$ to reduce the computational burden. We do not report the dropout rates since they are set to be the same as those in Gu et al. (2022a). Also, we made $\beta$ a trainable parameter.

## I  SUPPLEMENTARY EXPERIMENTS

### I.1  PREDICT THE MAGNITUDES OF WAVES

In Figure 1, we see an example of the frequency bias of SSMs, where the model is better at extracting the wave information of a low-frequency wave than a high-frequency one. In this section, we produce more examples on the same task to show that one is able to tune frequency bias by playing with $\alpha$ and $\beta$ we introduced in section 5.1 and 5.2, respectively.

| Task | Depth | #Features | Norm | Prenorm | $\alpha$ | LR | BS | Epochs | WD | $\Delta$ Range |
|------|-------|-----------|------|---------|----------|-----|-----|--------|-----|----------------|
| ListOps | 8 | 256 | BN | False | 3 | 0.002 | 50 | 80 | 0.05 | (1e-3,1e0) |
| Text | 6 | 256 | BN | True | 5 | 0.01 | 32 | 300 | 0.05 | (1e-3,1e-1) |
| Retrieval | 6 | 128 | BN | True | 3 | 0.004 | 64 | 40 | 0.03 | (1e-3,1e-1) |
| Image | 6 | 512 | LN | False | 3 | 0.01 | 50 | 1000 | 0.01 | (1e-3,1e-1) |
| Pathfinder | 6 | 256 | BN | True | 3 | 0.004 | 64 | 300 | 0.03 | (1e-3,1e-1) |
| Path-X | 6 | 128 | BN | True | 5 | 0.001 | 20 | 80 | 0.03 | (1e-4,1e-1) |

**Table 4:** Configurations of our S4D model, where LR, BS, and WD stand for learning rate, batch size, and weight decay, respectively. The hyperparameter $\alpha$ is the scaling factor introduced in section 5.1. We set the parameter in section 5.2 as a trainable parameter to reduce the need for hyperparameter tuning.

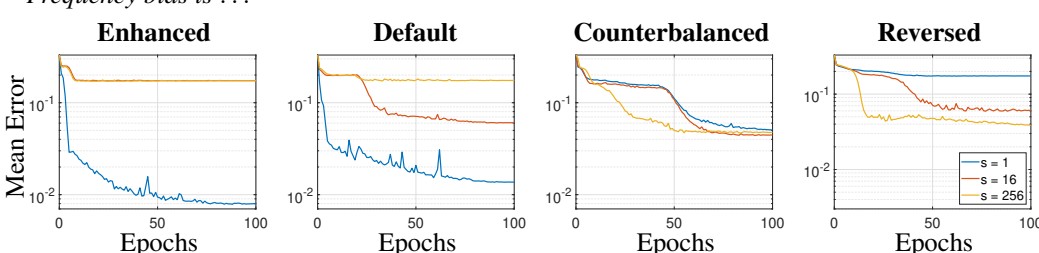

**Figure 10:** Reproduction of the experiment shown in Figure 1, with difference choices of $(\alpha, \beta)$. From the left to right, our choices of $(\alpha, \beta)$ are $(0.01, -1)$, $(1, 0)$ (the default), $(10, 0.5)$, and $(100, 1)$, respectively. The legend applies to all pictures. We can see that these choices allow us to enhance, counterbalance, or even reverse the frequency bias.

We see from Figure 10 that by tuning hyperparameters, we can change the frequency bias of an SSM. In particular, when $\alpha = 100$ and $\beta = 1$, we reversed the frequency bias so that the magnitude of the low-frequency wave $\cos(t)$ cannot be well-predicted, while a high-frequency wave is captured relatively well.

## I.2    TUNING FREQUENCY BIAS IN MOVING MNIST VIDEO PREDICTION

In this section, we present an experiment to tune frequency bias in a video prediction task. We show that our frequency bias analysis and the tuning strategies not only work for vanilla SSMs but also their variants. We examine a model architecture called ConvS5 that combines SSMs and spatial convolution (Smith et al., 2024). We apply the model to predict movies from the Moving MNIST dataset (Srivastava et al., 2015). In this dataset, two (or more) digits taken from the MNIST dataset (Deng, 2012) move on a larger canvas and bounce when touching the border. This forms a video over time. In our experiment, we slightly modify the movies by coloring the two digits. In particular, every movie contains two moving digits — a fast-moving red one and a slow-moving blue one. The speed of the red digit is ten times that of the blue digit; consequently, the red digit can be considered as a "high-frequency" component, whereas the blue digit is a "low-frequency" component. Our goal in this experiment is to use a ConvS5 model to generate up to 100 frames, conditioned on 500 frames. The ConvS5 model applies LTI systems to the time domain (i.e., the axis of the frames), but in the meantime incorporates spatial convolutions in the LTI systems, where the LTI systems are still initialized by the HiPPO initialization.

In this experiment, we train two models using two different initializations. The first initialization we use is the default HiPPO initialization. Then, we try another initialization, where for every $w_j$ that is the imaginary part of an eigenvalue of $\mathbf{A}$, we transform $w_j$ by

$$w_j \mapsto \text{sign}(w_j)(|w_j| + 200). \tag{14}$$

That is, we shift every $w_j$ away from the origin by 200. This does not correspond to any $\alpha > 0$ that we introduced in section 5.1, but our intuition is still based on our discussions in section 3 and 4. That is, when we move away every $w_j$ that is contained in $[-200, 200]$, our model is incapable

of handling the low frequencies. This is indeed observed in Figure 11: when we use the original HiPPO initialization, the high-frequency red digit cannot be predicted, whereas when we modify the initialization based on eq. (14), we well-predicted the red digit but the low-frequency blue digit is completely distorted.

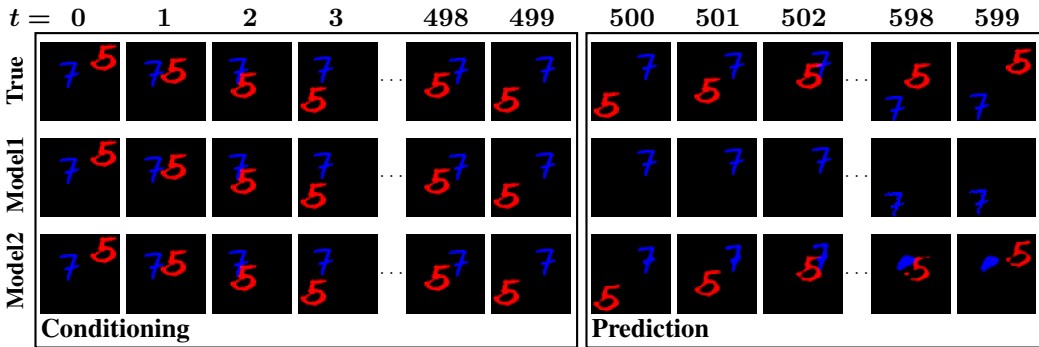

**Figure 11:** A ConvS5 model is trained to predict the Moving MNIST videos. In "Model1", we use the original HiPPO initialization; in "Model2", we modify the initialization based on eq. (14). We see that when we use the HiPPO initialization, only the slow-moving blue digit can be generated; on the other hand, pushing all eigenvalues of **A** to the high-frequency regions (see eq. (14)) allows us to predict the fast-moving red digit.

