# OpenReview forum: "Tuning Frequency Bias of State Space Models"
_ICLR.cc/2025/Conference — ICLR 2025 Spotlight_

### Official Review · Reviewer_tRCa · 2024-10-25

**Soundness:** 4
**Presentation:** 3
**Contribution:** 3
**Rating:** 8
**Confidence:** 2

**Summary:**

This paper addresses the frequency bias of State Space Models (SSMs) and provides methods to adjust this bias. The authors introduce a formal framework to characterize the frequency behavior of SSMs and reveal that standard SSMs, as commonly described in recent literature, tend to learn low frequencies. While this low-frequency bias is often advantageous, the authors argue that it is possible to improve the model's frequency response.

To achieve this, they propose two different approaches: the first modifies initialization to control which frequencies the model can learn and the second alters the transfer function to adjust how much the model emphasizes particular frequencies.

Although I am not a expert in SSMs, I found the paper well-structured and effective in explaining complex issues with clear, illustrative examples.

**Strengths:**

Well-written with effective illustrative explanations.

Provides a theoretically sound analysis of a complex problem.

Practical relevance is high.

**Weaknesses:**

The analysis is limited to diagonal systems.

Tuning the additional hyperparameters, α and β, may pose challenges in practice.

**Questions:**

Could you provide some guidelines for tuning the α and β parameters? How much tuning was necessary to achieve the results in Table 3?

How could the analysis be extended to non-diagonal systems?

---

> ### Author Response · Authors · 2024-11-19
> **Response to the Reviewer**
>
> We thank the reviewer for the careful review and insightful comments. Below we address the questions and comments raised in this review.
>
> * Limitation to diagonal systems: We acknowledge that our analysis of the training and the proposed hyperparameter $\alpha$ is indeed restricted to the diagonal setting. On the other hand, the Sobolev-norm-based filter is valid for any SSM. We would also like to highlight the fact that most modern SSMs that are used in practice rely on a diagonal matrix $\mathbf{A}$, so our methods still enjoy wide applicability. Extending the analyses to the nondiagonal case is an interesting problem and can probably be achieved by some control mechanisms or spectral theory (see [1] and [2] for some example analyses not in the context of frequency bias). That is, as long as $\mathbf{A} = \mathbf{V} \boldsymbol{\Lambda} \mathbf{V}^{-1}$ is diagonalizable, the systems $(\mathbf{A}, \mathbf{B}, \mathbf{C}, \mathbf{D})$ and $(\boldsymbol{\Lambda}, \mathbf{V}^{-1}\mathbf{B}, \mathbf{C}\mathbf{V}, \mathbf{D})$ are equivalent. Hence, frequency bias can still be tuned as long as one can control the eigenvalues of the nondiagonal matrix $\mathbf{A}$.
>
> * Tuning hyperparameters: In our experiments, we have set $\beta$ to be a trainable parameter, which does not require any tuning. We have provided a more concrete guideline for tuning the hyperparameter $\alpha$ in Appendix G. We have done relatively little tuning when doing the experiments in Table 3, only trying 2-3 different values of $\alpha$ for each task.
>
> We hope this answers the reviewer's questions and concerns. We are happy to answer any follow-up question(s) that the reviewer may have.
>
> [1] Erichson, N. B., Azencot, O., Queiruga, A., Hodgkinson, L., Mahoney, M. W.,  Lipschitz recurrent neural networks, International Conference on Learning Representations, 2021.
>
> [2] Yu, A., Nigmetov, A., Morozov, D., Mahoney, M. W., Erichson, N. B., Robustifying state-space models for long sequences via approximate diagonalization, International Conference on Learning Representations, 2023.

---

> > ### Comment · Reviewer_tRCa · 2024-11-22
> > **Thank you for your reply**
> >
> > Thank you for your reply. I will stick to my recommendation of acceptance.

---

### Official Review · Reviewer_uuaF · 2024-10-30

**Soundness:** 3
**Presentation:** 4
**Contribution:** 3
**Rating:** 8
**Confidence:** 4

**Summary:**

This paper studies the frequency bias of state space models, where the authors identify a tendency, under generic initialisations of the state space matrix, to favor the response to low frequency signals. The authors give a definition of the frequency bias through the total variation of the transfer function, in that a low TV over an interval in frequency space is understood as a bias to not capture this region effectively. The authors then show that generic initialisations indeed gives lower TV for high frequency regions, and moreover that this cannot be fixed by optimisation (in the sense that the gradient on the frequency component is small if there is little initial overlap).

Two solutions are proposed to mitigate this issue: better initialisation where a hyper-parameter is introduced to weigh higher frequencies differently, or changing the loss function by adding a weighting function in frequency space to promote training larger frequencies (or vice versa). Improvements are observed on both a synthetic denoising task and benchmarks from long range arena.

**Strengths:**

The paper is very clearly written, and the motivation and implications of the theoretical results are clearly explained. The identification of the frequency bias is useful for problems where high frequency components need to be extracted. The proposed methods to mitigate the frequency bias appear simple to implement, and are principled.

**Weaknesses:**

1. The paper can be improved by clarifying some notational and definition issues, and these are detailed in the questions below.
2. Notational suggestion: the notation of using $y_j$ to denote imaginary part of the diagonal of $A$ can be changed to avoid confusion with the output of the network, also called $y$.
3. Since the SSM transfer function is differentiable, it may be simpler to define the total variation as the integral of its derivatives to avoid unncessary complications.

**Questions:**

1. Can the authors explain more clearly why the rate of change of the transfer function is a proper measure of frequency bias? Traditionally (e.g. low pass filters) would consider the support of the transfer function, or its magnitude over frequency intervals, as a measure of its response to frequency components. Some proper definition of "frequency sensitivity" would make the definition of frequency bias in terms of TV more transparent.
2. After proposition 1, Rule II: what happens in the limit $L\to\infty$?
3. Eq 7: I find it strange that the weighting factor is $(1+|s|)^{2\beta}$ instead of $(1+|s|^2)^{\beta}$, as in the usual Bessel potential spaces. What's the motivation for this departure? In any case, the name "the Sobolev norm" is unclear since there are many Sobolev spaces.
4. For the mitigation algorithms, what are some general tuning strategies for $\alpha$ and $\beta$? Fig 5 seems to say $\alpha$ is more sensitive than $\beta$. Some comments on this would be helpful.

---

> ### Author Response · Authors · 2024-11-19
> **Response to the Reviewer**
>
> We thank the reviewer for the careful review and insightful comments. Below we address the questions and comments raised in this review.
>
> * W1: Please find our detailed response to the questions.
>
> * W2: Thank you for pointing out the issue with notation overlap. We have changed $a = x + iy$ into $a = v + iw$, where we refrained from using $a = u + iv$ since $\mathbf{u}$ is reserved for the inputs of the LTI systems.
>
> * W3: We have added the integral definition of the total variation to our manuscript.
>
> * Q1: The reviewer has correctly identified that a high-magnitude transfer function is sufficient for a frequency to "pass." Here, we provide more intuitions in terms of why the total variation is a good measurement of the frequency bias. As noted in [2], one reason why an LTI system in an SSM degenerates is that its transfer function is too flat. That is, if $\mathbf{G}$ can be well-approximated by a much lower-degree rational function, then one can essentially replace the large LTI system with a much smaller one. In other words, our LTI system is not as expressive as its size may have suggested. If we now restrict our attention to only a part of the frequency domain, then the same reasoning applies: if the transfer function $\mathbf{G}$ is flat on $[a,b]$, then we can replace a large LTI system with a much smaller one. While the small system may not well-approximate the original system on the entire frequency domain, the approximation is good on $[a,b]$. Hence, this essentially means that our LTI system is unable to capture "complex patterns" in the frequency interval $[a,b]$ because all it does in $[a,b]$ can also be done by a much smaller system. We have added a similar discussion of this in Appendix A.
>
> * Q2: As $L \rightarrow \infty$, $\tan((1 - (L-1)/L) \pi / 2) \rightarrow \infty$ so the "edge of nonzero information" could grow arbitrarily large. However, this is only due to the fact that very few samplers of the transfer function $\mathbf{G}$ will become large. See Figure 7 (Right), for example. While there exists a sampler whose imaginary part $> 10000$, most are actually $< 1000$. That being said, as $L \rightarrow \infty$, one can always make up a synthetic input data of which $\alpha$ needs to be arbitrarily large to capture. However, in the average case, the few large samplers would only contain a vanishing proportion of the information as $L \rightarrow \infty$ --- that is why Rule I does not involve $L$.
>
> * Q3: The reviewer raised a good point. Different choices are made in the definition of the Sobolev norm. These norms are different but norm-equivalent (i.e., $c \\|\cdot\\|_x \leq \\|\cdot\\|_y \leq C \\|\cdot\\|_x$ for constants $c, C > 0$) so that they lead to the same Sobolev space. In our case, the $(1+|s|^2)^\beta$ and $(1+|s|)^{2\beta}$ factors lead to equivalent norms, although they are numerically different. The factor we chose came from the spherical harmonics community [1], but we have remarked that in our manuscript to avoid potential confusion.
>
> * Q4: We have added a concrete guideline for tuning $\alpha$ in Appendix G. For the LRA tasks, we do not set $\beta$ as a tuning parameter but instead make it a trainable parameter. In general, we find that on LRA tasks, tuning $\alpha$ is indeed more effective than training $\beta$. However, our ablation study below still shows that the model benefits from having a $\beta$ involved.
>
>    |         | ListOps | Text | Retrieval  | Image | PathFinder | PathX |
>    |:-------:|:-------:|:-------:|:-------:|:-------:|:-------:|:-------:|
>    |only $\alpha$|61.24|89.32|91.35|90.51|95.74|97.52|
>    |only $\beta$|61.15|88.92|90.31|89.75|95.93|96.12|
>    |with neither|60.47|86.18|89.46|88.19|93.06|91.95|
>    |with both|62.75|89.76|92.45|90.89|95.89|97.84|
>
>    Moreover, for some tasks where one needs a more extreme frequency bias, having only $\alpha$ is suboptimal, while $\beta$ is really making a big difference. (See Table 2 for example.)
>
> We hope this answers the reviewer's questions and concerns. We are happy to answer any follow-up question(s) that the reviewer may have.
>
> [1] J. A. Barcelo, M. Folch-Gabayet, T. Luque, S Perez-Esteva, and M. C. Vilela. The Fourier extension operator of distributions in Sobolev spaces of the sphere and the Helmholtz equation. Proc. Roy. Soc. Edin. Sec. A: Math., 151(6):1768–1789, 2021.
>
> [2] A. Yu, M. W. Mahoney, N. B. Erichson, HOPE for a robust parameterization of long-memory state space models, arXiv preprint arXiv:2405.13975, 2024.

---

> > ### Comment · Reviewer_uuaF · 2024-11-21
> > **Questions addressed**
> >
> > Thank you for addressing most of my questions. I think this is a good contribution.
> >
> > For your explanation to Q1, one should be more cautious, because exponential kernels induced by LTI systems cannot in general be modified locally without affecting its effect globally. In particular, the argument "if $\mathbf{G}$ can be well-approximated by a much lower-degree rational function, then one can essentially replace the large LTI system with a much smaller one" is true globally in time, but *not* locally - a counter example would be a (linear combination of) delta response function, which is a.e. flat, but this does not mean that it can be approximated by small LTI systems. The statement is true for convolution models where you can change the response function locally independently of other regions.

---

> > > ### Author Response · Authors · 2024-11-21
> > > **Thank you for your reply**
> > >
> > > We thank the reviewer for carefully reading through our response and confirming that the questions have been addressed. In response to the potential confusion in our explanation to Q1, we have updated Appendix A of our manuscript:
> > >
> > > 1. We removed vague terms such as "replace an LTI system with a smaller one." Instead, we formally say what we mean by "approximating a large LTI system by a smaller one."
> > >
> > > 2. When discussing the transfer function on the interval $[a,b]$, we say a flat transfer function on $[a,b]$ indicates that "there exists a much smaller LTI system $\tilde{\Gamma}$ and its actions on the Fourier modes in $[a,b]$ are very similar to those of the original system." Hence, we emphasize that our reasoning is *local in the frequency domain* (and global in the time domain).
> > >
> > > We hope this improves the clarity of our explanation. Thank you again for your careful review!

---

### Official Review · Reviewer_XLdv · 2024-10-31

**Soundness:** 3
**Presentation:** 3
**Contribution:** 3
**Rating:** 6
**Confidence:** 3

**Summary:**

In this paper, the authors find that when training state space models with canonical initialization methods, there is an implicit bias such that the models are implicitly toward to capture low-frequency components instead of high-frequency ones.
To encourage the models to capture a more broader frequency pattern, the authors propose two mechanisms: one is scaling the initialization and another method is to apply a Sobolev-norm-based filter on the state space models.
By tuning the frequency bias, the state space models are shown to be able to have better performance on the long range arena benchmark.

**Strengths:**

1. The organization and presentation of this paper is smooth and clear, and it provides a better understanding on the training mechanisms of state space models on sequence modeling.

2. I find the paper to be well-written and easy to follow. The overall topic of state space models is important.

**Weaknesses:**

1. In the end of Section 2, the authors state that the larger the total variation of the transfer function is, the better an LTI system is at distinguishing the Fourier modes. This claim is not intuitive for me, and the total variation cannot distinguish the following two different cases: the first one is large amplitude with low frequency and the second one is small amplitude with high frequency. For example, $G$ is an impulse response vs $G$ is a sinusoidal wave with a small amplitude. From my understanding, these two LTI systems have different ability on distinguishing the Fourier modes.

2. The statement after Lemma 1 that small $|y_i|$ induces small total variation seems to be wrong. From the upper bounds in Lemma 1, if $|y_i|$ decreases, the the upper bounds increase, which means that the total variation will be large.

3. The initialization method in Corollary1 is not a commonly used method. For S4D, the initialization methods for $a$ are mainly S4D-Legs and S4D-Lin. Why not choose these two initialization methods instead?

**Questions:**

For the numerical experiments (Table 3) on the long range arena benchmark, how is the $A$ matrix initialized? Is $A$ initialized by the method in Corollary 1? There are some ablation studies on the scaling of the imaginary part of $A$ in the S4D paper [1]. It is shown in [1] that scaling all imaginary parts by a constant factor substantially reduce the performance of S4D, so I am curious that if we only scaling the imaginary part of $a$, does it help to improve the performance on the long range arena benchmark? It would be more convincing if the authors can provide more experiment results of only scale $a$, only train $\beta$, the results in Table 3 is the case for scale $a$ + train $\beta$.






[1] Gu, Albert, et al. "On the parameterization and initialization of diagonal state space models." Advances in Neural Information Processing Systems 35 (2022): 35971-35983.

---

> ### Author Response · Authors · 2024-11-19
> **Response to the Reviewer**
>
> We thank the reviewer for the careful review and insightful comments. Below we address the questions and comments raised in this review.
>
> * W1: When we say "distinguishing," we have in mind two input signals that pass through the same LTI system. If an LTI system is good at "distinguishing" the low-frequency signals, then that means if we have two different low-frequency signals, say $\cos(t)$ and $\cos(1.1 t)$, then the two corresponding outputs would be very different: for instance, $0.1\cos(t)$ and $10\cos(1.1 t)$. We believe that another source of confusion is the conversion between the time domain and the frequency domain. The transfer function $\mathbf{G}$ is always a rational function and is defined on the frequency domain. For example, given an input signal $t \mapsto \cos(kt)$ for a fixed $k$, the output is (ignoring the constants in the Fourier transform)
> \\[
>     \mathbf{y}(t) = (\mathbf{G}(is) (\cos(kt))^\wedge)^\vee(t) = (\mathbf{G}(is) (\delta\_{-k}(s) + \delta_{k}(s)))^\vee(t) = (\mathbf{G}(-ik) + \mathbf{G}(ik)) \cos(kt),
> \\]
> where $\delta$ is the Dirac delta function. Hence, if $\mathbf{G}(ik)$ has a large total variation when $k$ is close to zero, then when $k$ varies in the low-frequency region (i.e., when $|k|$ is small), the output given the low-frequency input $\cos(kt)$ will change rapidly. This is how frequency bias is related to the total variation of the transfer function. We have added Figure 6 in our manuscript to further clarify this and we have also explained below why this is a good definition of frequency bias. We are happy to answer any follow-up questions from the reviewer.
>
> * W2: When discussing this lemma, we have in mind that $B \rightarrow \infty$ while $y_j$'s are unchanged. That is, we fix an initialization and let the bandlimit $B$ increase. Then, as $B$ gets larger, i.e., when the frequencies get higher, the total variation would vanish eventually. We have reworded this statement in our manuscript.
>
> * W3: The initialization studied in Corollary 1 is indeed the S4D-Lin initialization (see Listing 1 of [1] and the implementation in the original $\texttt{s4}$ repository: https://github.com/state-spaces/s4/blob/main/models/s4/s4d.py by Gu et al.) We have also added a result for the HiPPO-LegS initialization (see Corollary 2 in Appendix B).
>
> * Question from the reviewer: We confirm that we scale the initialization by multiplying the imaginary parts of each diagonal entry of $\mathbf{A}$ by a hyperparameter $\alpha$. The ablation study in [1] considers only multiplying the imaginary parts by $0.01$ and $100$, i.e. $\alpha = 0.01$ or $100$. These numbers are a bit too extreme, and from Figure 5 in our manuscript, we can see that a substantially reduced accuracy is well expected. In our case, we normally set $\alpha$ to be $3$ or $5$ when learning the LRA tasks, which gives a better amount of frequency bias. We provide in Figure 5 and section 6 (III) an ablation study where we only changed $\alpha$ and only changed $\beta$. In the meantime, we are happy to provide more comprehensive data on LRA tasks:
>
>    |         | ListOps | Text | Retrieval  | Image | PathFinder | PathX |
>    |:-------:|:-------:|:-------:|:-------:|:-------:|:-------:|:-------:|
>    |only $\alpha$|61.24|89.32|91.35|90.51|95.74|97.52|
>    |only $\beta$|61.15|88.92|90.31|89.75|95.93|96.12|
>    |with neither|60.47|86.18|89.46|88.19|93.06|91.95|
>    |with both|62.75|89.76|92.45|90.89|95.89|97.84|
>
>    Right now, the rows with "only $\alpha$" and "only $\beta$" are done with one random seed. We will update the manuscript with results from multiple trials once they finish
>
> We hope this answers the reviewer's questions and concerns. We are happy to answer any follow-up question(s) that the reviewer may have.
>
>  [1] Gu, Albert, et al. "On the parameterization and initialization of diagonal state space models." Advances in Neural Information Processing Systems 35 (2022): 35971-35983.

---

> > ### Comment · Reviewer_XLdv · 2024-11-25
> > **Thank you for your reply**
> >
> > Thank you for your reply, I recommend acceptance of this paper.

---

### Official Review · Reviewer_joFK · 2024-11-04

**Soundness:** 4
**Presentation:** 4
**Contribution:** 4
**Rating:** 8
**Confidence:** 3

**Summary:**

This paper studies the frequency bias for State-space-models (SSMs). With strategies to select better initialization and Sobolev filtering (during training), this problem could be mitigated. In the experimental study, this manuscript also provides extensive results to show the effectiveness of their proposed strategies.

**Strengths:**

This paper presents a remarkable analysis of the frequency bias in state-space models (SSMs). It has been commonly observed that neural networks tend to fit low-frequency information first, often filtering out high-frequency data in many cases. This paper critically emphasizes the importance of initialization and introduces a strategy to mitigate this problem using a Sobolev filter. The figures presented in the experiments are highly inspiring. As a theoretical work, this paper provides sufficient experimental results, demonstrating strong performance.

**Weaknesses:**

The weaknesses of this paper mostly come from two parts.

1. The analysis is based on SSM which is potentially hard to be generalized on general neural networks.

2. Even though the reason is getting clearer, these strategies of this paper could still be suboptimal for practitioners.

**Questions:**

1. The authors may need to define HiPPO in line 92 Page 2 before mentioning it.

2. How will these discoveries help for general neural network settings?

3. Would the Sobolev filter difficult to compute in practice (when doing the SSM training)?

4. How to interpret Figure 4 when having low-freq. noise and high-freq. noise. What is the difference? Additionally, why low-freq. noise can be denoised (with higher-frequency-pass filter) but the images in the bottom rows will be reconstructed?

---

> ### Author Response · Authors · 2024-11-19
> **Response to the Reviewer**
>
> We thank the reviewer for the careful review and insightful comments. Below we address the questions and comments raised in this review.
>
> * Generalization to general NNs: We acknowledge that many discussions in this paper are limited to the SSM case. The scaling of the initialization is not applicable to a general neural network. The Sobolev-norm-based ideas, on the other hand, can be extended to the general neural network case, where the one-dimensional Fourier modes are extended to the multi-dimensional spherical harmonics. Some related works are discussed in our introduction.
>
> * Suboptimality for practitioners: We admit that the introduction of the additional hyperparameter leads to more work for the practitioners. We have set $\beta$ to be a trainable parameter so that it would not require additional tuning. Moreover, we have added more concrete guidelines for a practitioner in Appendix G to make the tuning of $\alpha$ much easier.
>
> * Q1: Thank you for pointing out this issue. We have explained what HiPPO is when we first discuss it on line 92.
>
> * Q2: Please see our discussion above for a general NN.
>
> * Q3: There are several ways to compute the Sobolev filter. If an SSM is trained in the frequency domain, then applying the Sobolev filter reduces to multiplying the FFT of the convolutional kernel by the corresponding scaling factors, which costs almost nothing. If an SSM is trained using scan algorithms, then we propose to preprocess the inputs before applying the standard scan algorithm, because one can alternatively view the Sobolev filter as one applying to the inputs instead of the LTI systems. We have included more details in Appendix F.
>
> * Q4: In Figure 4, there are two types of noises: one with the horizontal stripes and the other with the vertical stripes. We flatten the images in the row-major order. That is, we form the sequence by grabbing the pixels from the first row, then the second row, then the third, etc. With the horizontal stripes, the noise does not change within a row, resulting in a low-frequency noisy sequence. With the vertical stripes, the noise changes rapidly within a row, resulting in a high-frequency noisy sequence. We provided Figure 9 for a visual demonstration. Hence, when a high-pass filter is applied, the low-frequency noises are blocked (second row, last image) while the high-frequency noises still pass (last row, last image). When a medium-to-low-pass filter is applied, on the other hand, we can see that the low-frequency noises pass (second row, second-last image) whereas the high-frequency noises are blocked (last row, second-last image).
>
> We hope this answers the reviewer's questions and concerns. We are happy to answer any follow-up question(s) that the reviewer may have.

---

### Author Response · Authors · 2024-11-19
**Summary of the Rebuttal Revision**

We appreciate the reviewers' valuable feedback, which has helped improve this manuscript. Along with the minor changes detailed in our responses to each reviewer, we have compiled a list of key updates (highlighted in red in the rebuttal version) made during the review process:


* We have added Appendix A to further clarify our definition of the frequency bias of SSMs.

* We have added more analyses of the frequency bias of other popular initialization schemes in Appendix B.

* In Appendix F, we have discussed how to implement the Sobolev-norm-based filter with various SSM algorithms.

* Appendix G now includes more concrete and detailed guidelines for tuning the $\alpha$ hyperparameter.

* We have updated the notation for an eigenvalue $a$ of $\mathbf{A}$ from $a = x + iy$ to $a = v + iw$ throughout the paper for consistency.

---

### Author Response · Authors · 2024-11-25
**Thanks to the reviewers**

As the deadline for the rebuttal revision approaches, we would like to thank all reviewers again for their careful reviews, which have further improved this manuscript, and positive evaluations.

---

### Public Comment · ~Albert_Gu1 · 2025-03-07
**Question**

Cool paper! I had a brief question about the choice of initialization used. As I understand it, the paper primarily used the version of S4D with linear scaling on the imaginary parts ($A_k = -0.5 + i k \pi$) which has small imaginary parts and according to this paper, it has difficulty learning high-frequency components. Did the authors ever test the other canonical initialization based off of the original "LegS" initialization? That one has much higher imaginary parts, so I wonder whether it still has the same sort of biases?

---

> ### Public Comment · ~Annan_Yu1 · 2025-03-08
> **Thank you for your question!**
>
> Dear Albert,
>
> Thank you for your question! We are glad (and honored) that you found our paper interesting. We confirm that the prototype we studied in this work is the S4D-Lin framework. The question about the HiPPO-LegS initialization is certainly a fair and interesting one. While we have not empirically tested the frequency bias of HiPPO-LegS, we outline some lines of thoughts that could be useful:
>
> 1. If we consider an S4D model with the S4D-LegS or S4D-Inv initialization, which diagonalizes the rank-one perturbation of the HiPPO-LegS matrix $\mathbf{A}$, then there are indeed eigenvalues with much larger imaginary parts. In this case, we do expect that the model is better at capturing the high frequencies than one initialized by S4D-Lin. One thing to note is that the S4D-LegS (or S4D-Inv) initialization does not place the eigenvalues equispacedly in the Fourier domain. In particular, there are much fewer eigenvalues in the high-frequency domain than the low-frequency one. The frequency bias depends on both the support and the density of the spectrum of $\mathbf{A}$. That being said, in some cases where there is a high demand for learning high frequencies, scaling the S4D-Lin initialization into the high-frequency domain may be more effective than using S4D-LegS or S4D-Inv directly. However, S4D-LegS and S4D-Inv may exhibit an inductive bias tailored to other tasks, where one needs to learn just a little bit (but not too much) of the high frequencies. In general, we believe that frequency bias can be better tuned if one knows some spectral information about the problem at hand.
>
> 2. Still with the S4D model, another way to diagonalize the HiPPO-LegS matrix is to marginally perturb it and then diagonalize, which was studied in our earlier work (https://openreview.net/forum?id=DjeQ39QoLQ). In that case, we did witness a lot of eigenvalues with very large imaginary parts. The fact that this diagonalization strategy showed an improved performance can potentially be attributed to them. However, one thing that we would like to note is that in this initialization, when an eigenvalue has a large imaginary part, it is also usually accompanied by a very negative real part. We did not explicitly discuss the real parts in our paper, but as noted in several works (e.g., https://openreview.net/forum?id=SGwbNf4SQD, https://openreview.net/forum?id=RZwtbg3qYD, https://openreview.net/forum?id=sZJNkorXMk), a very negative real part in general harms the expressiveness of that specific hidden state. That means the frequency bias may not have been changed by as much as the imaginary parts of the eigenvalues suggest.
>
> 3. If we use a more stable decomposition of HiPPO-LegS, e.g., the diagonal-plus-rank-one S4 model, things become more complicated because one cannot directly read off the eigenvalues of $\mathbf{A}$. It is true that during training, the matrix $\mathbf{A}$ can almost surely be diagonalized (and therefore the transfer function can be written into partial fractions), but this partial fraction representation may be insanely unstable, leading to issues such as spurious poles and Froissart doublets. In that case, merely looking at the locations of the poles (i.e., eigenvalues of $\mathbf{A}$) can be very misleading (for example, the pseudospectral theory says that perturbing the HiPPO-LegS matrix $\mathbf{A}$ by a little bit would entirely change the spectrum of it, so the notion of eigenvalues itself is ill-posed), and the only way to assess the frequency bias is by analyzing the transfer function as a whole. That being said, when $\mathbf{A}$ is not parameterized by a diagonal matrix, our paper does not provide a direct way to analyze and tune the frequency bias. However, this seems like an interesting research direction that can have a broader impact.
>
> Please let us know if you have any comments or follow-up questions. We are more than happy to discuss them further!
>
> Annan (on behalf of the paper authors)

---

> > ### Public Comment · ~Annan_Yu1 · 2025-03-08
> > **Correction to Point 1**
> >
> > We have noticed that we negligently referred to an S4D-LegS formula online that misses a minus sign. Since the matrix is skew-symmetric instead of symmetric, we have updated our first point to correctly reflect our view. We apologize for any confusion our earlier post may have caused. We hope the updated message helps!

---

### Meta-Review · Area_Chair_CGNJ · 2024-12-23

**Metareview:**

The authors are concerned with the observation that state space models contain an implicit bias for ignoring high-frequency components. After conducting a theoretical analysis to shed light into this phenomenon, they formulate a theory (related to initialization) which, in turn, gives rise to their two proposed solutions. This is, therefore, an overall well-conducted piece of research, from motivation to solution proposal, which is a great addition to the literature.

**Additional Comments On Reviewer Discussion:**

There were two key discussion points which I think are the most relevant:

1. Generality of the method. Firstly, the results can be suboptimal for practitioners due to the introduction of extra parameters. Secondly, the analysis is mostly focused on SSMs, as opposed to general neural networks. The discussion has clarified these concerns in the following way: firstly, the authors overall engage in a very honest and constructive way, acknowledging the limitations while pointing out how these are not so severe after all. For example, they give recipes for training hyperparameters as well as pointing out that the Sobolev-norm-based ideas can be extended to the general neural network case.

2. Various clarifications were requested, which the authors addressed adequately.

---

### Decision · Program_Chairs · 2025-01-22

Accept (Spotlight)